# The vertebrate Aqp14 water channel is a neuropeptide-regulated polytransporter

François Chauvigné[1], Ozlem Yilmaz[2], Alba Ferré[1], Per Gunnar Fjelldal[3], Roderick Nigel Finn [1,2]* & Joan Cerdà [1]*

Water channels (aquaporins) were originally discovered in mammals with fourteen sub-families now identified (AQP0-13). Here we show that a functional Aqp14 subfamily phylogenetically related to AQP4-type channels exists in all vertebrate lineages except hagfishes and eutherian mammals. In contrast to the water-selective classical aquaporins, which have four aromatic-arginine constriction residues, Aqp14 proteins present five non-aromatic constriction residues and facilitate the permeation of water, urea, ammonia, $H_2O_2$ and glycerol. Immunocytochemical assays suggest that Aqp14 channels play important osmoregulatory roles in piscine seawater adaptation. Our data indicate that Aqp14 intracellular trafficking is tightly regulated by the vasotocinergic/isotocinergic neuropeptide and receptor systems, whereby protein kinase C and A transduction pathways phosphorylate highly conserved C-terminal residues to control channel plasma membrane insertion. The neuropeptide regulation of Aqp14 channels thus predates the vasotocin/vasopressin regulation of AQP2-5-6 orthologs observed in tetrapods. These findings demonstrate that vertebrate Aqp14 channels represent an ancient subfamily of neuropeptide-regulated polytransporters.

[1] IRTA-Institute of Biotechnology and Biomedicine (IBB), Universitat Autònoma de Barcelona, 08193 Bellaterra, (Cerdanyola del Vallès), Spain. [2] Department of Biological Sciences, Bergen High Technology Centre, University of Bergen, 5020 Bergen, Norway. [3] Institute of Marine Research, Matre Aquaculture Research Station, 5984 Matredal, Norway. *email: nigel.finn@uib.no; joan.cerda@irta.cat

The aquaporin superfamily of membrane proteins was first discovered almost 30 years ago through demonstration of the biophysical properties of human AQP1 in frog oocytes[1]. The discovery validated the long search for molecular channels that evolved to specifically facilitate the rapid yet selective transport of water across biological membranes[2]. With the subsequent discovery of other aquaporins, structural studies have revealed overall similarities of the monomeric proteins. This includes intracellular N- and C-termini, often involved in cytoplasmic protein trafficking, 6 transmembrane domains (TMD) linked by loops A–E, and two hemihelices that fold to oppose canonical Asp-Pro-Ala (NPA) motifs in the outer vestibules. In addition, four aromatic-arginine (ar/R) residues on TMD2, −5 and loop E typically form important selectivity filters surrounding the central channel[3–5]. Each transmembrane protein functions in tetrameric assemblages to facilitate the passive single-file conductance of water or the other small uncharged solutes down their concentration gradients[6].

In the ~3 decades of research following the discovery of aquaporins, eutherian mammalian water channels have been classified according to the chronology of gene discovery, with 13 subfamilies (AQP0 – 12) initially identified[2,7,8]. Broader phylogenetic studies in diverse ranges of eukaryotic organisms have supported this view for vertebrates[9–14]. In a recent study of the evolution of this superfamily, however, we phylogenetically identified three entirely novel subfamilies (aqp14, −15 and −16) in vertebrates, and further showed that the AQP13 aquaporin subfamily, first documented as AQPxlo in Xenopus laevis oocytes[15], exists as a complete ortholog in Amphibia, but also as a pseudogene in the prototherian order of egg-laying mammals, the Monotremata, while aqp15 and aqp16 are restricted to specific lineages of fishes, turtles and crocodylians[16]. The vertebrate aquaporin superfamily is thus currently comprised of 17 subfamilies (AQP0 – 16) that are phylogenetically classified into four grades that can be traced to basal metazoan or parazoan lineages including Cnidaria (jelly fish and corals) or Porifera (sponges). This includes: classical aquaporins (AQP0, −1, −2, −4, −5, −6, −14 and 15) that primarily transport water, Aqp8-type aquaporins (AQP8 and −16) that primarily transport water, urea, ammonia and peroxide, aquaglyceroporins (AQP3, −7, −9, 10 and −13) that primarily function as water, urea and polyol transporters, and the intracellular unorthodox aquaporins (AQP11 and −12), for which functional data are mostly lacking, except for AQP11 which seems to transport water and glycerol[17–20].

The physiological roles of the different channels have been best studied in eutherian mammals, with major roles demonstrated in vision (AQP0), erythrocyte volume regulation (AQP1), vasopressin-regulated antidiuresis (AQP2), transcellular fluid transport and skin hydration (AQP3), the blood–brain barrier (AQP4), sweat and tear production (AQP5) and adipocyte metabolism (AQP7)[8]. Studies of non-mammalian vertebrates have also revealed the physiological importance of AQP1 and AQP8 orthologs in the germ cell biology and osmoregulation of fishes[21–26], and AQP2, −3, −5 and −6 orthologs in the water conservation of amphibians[27]. To date, however, no functional data exist for the novel aqp14 gene subfamily, which has no specific annotation in available genome databases, yet is suggested to exist in a broad range of vertebrates[16].

To understand the genomic landscape and function of the novel aqp14 water channel subfamily, we focused on piscine genomes, which remain the least annotated, yet encode proteins that span >500 million years of evolution, and which represent species that have adapted to freshwater and marine environments. The homeosmotic biology of fishes that live in these opposing osmotic envirmonments is fundamentally different, where the physiological task in freshwater species is to keep water out due to the hyperosmotic condition of their body fluids, while

that of marine fishes is the reverse. These latter species, whose blood osmolality is about 1/4 that of seawater, need to obtain pure water from the dessicating saltwater environment. Within true bony fishes (Osteichthyes), the evolved solution amongst teleosts resulted in every species drinking seawater and managing the desalination and water transport of the imbibed fluid along the length of their intestines, while secondarily excreting excess salts from chloride cells in the gill[28]. We therefore investigated the potential of Aqp14 proteins to function in fish osmoregulation, and here provide a comprehensive overview of the channel history by leveraging 190 and 87 piscine genomes and transcriptomes, respectively, to assemble >1000 exons into 179 full-length and 26 partial aqp14 coding sequences (CDS). This approach allowed us to identify lineage-specific pseudogenes and to reveal the origin, evolution and structure of the subfamily. We further experimentally demonstrate the molecular function and neuropeptide regulation of the Aqp14 channel from ancient and modern lineages of fishes, and confirm the existence of the complete Aqp14 ortholog in all extant sarcopterygian lineages, except eutherian mammals.

## Results

**Phylogeny.** To illustrate the phylogenetic divisions of the major aquaporin subfamilies in vertebrates, we initially assembled the complete set of full-length transcripts from the prototherian platypus (Ornithorhynchus anatinus) and analysed them together with the established genomic repertoire of the zebrafish (Danio rerio)[29]. We selected the platypus, since we have previously shown that its genome also retains the AQP13 ortholog found in Amphibia, and each of the other subfamilies reported for tetrapods, except AQP10. Thus, with the exception of AQP16, which is an AQP8-type channel identified in amphibians, turtles and crocodylians, the platypus and zebrafish display each of the major divisions of aquaporin subfamilies (AQP0-15) that are representative for the two major lineages of osteichthyan vertebrates (Sarcopterygii and Actinopterygii) with the highest aquaporin gene copy numbers[16,19,20,29]. Since Ensembl (v96) only lists four complete aquaporins for the platypus, we assembled the full CDS de novo from the genome based upon the manual identification of 67 exons (see materials and methods). This yielded 14 paralogs in the platypus compared to 19 in zebrafish. Phylogenetic analysis of the aligned codons shows that the platypus genome encodes seven classical aquaporins (AQP0, −1, −2, −4, −5, −6 and −14), one Aqp8-type channel (AQP8), two unorthodox paralogs (AQP11 and −12) and four aquaglyceroporins (AQP3, −7, −9 and −13) (Fig. 1a). We did not detect AQP10 in the platypus genome, and therefore included the metatherian Tasmanian devil (Sarcophilus harrisii) AQP10 ortholog for comparison with the zebrafish repertoire. As previously shown, the zebrafish genome encodes single-copy or duplicated orthologs of all of the mammalian channels, except for AQP2, −5, −6 and −13, which are not found in the genomes of ray-finned (actinopterygian) or cartilaginous (chondrichthyan) fishes[16,29,30]. In contrast to prototherian mammals, the zebrafish encodes an Aqp15 channel, which is also present in the genomes of other actinopterygian or chondrichthyan fishes[16,26]. This analysis shows the predominance of the classical-type aquaporins and aquaglyceroporins in mammalian and piscine genomes, with AQP14 clustering next to AQP4 within the classical grade of aquaporins.

We then identified and assembled 205 Aqp14 proteins and their respective CDS from 122 families within 59 orders of fishes encompassing ~68% of the known orders of actinopterygian fishes[31]. Bayesian phylogenetic analyses (25 million MCMC generations) of the aligned codons without exon 1, which encodes the 3–7 variable N-terminal amino acids, provided a

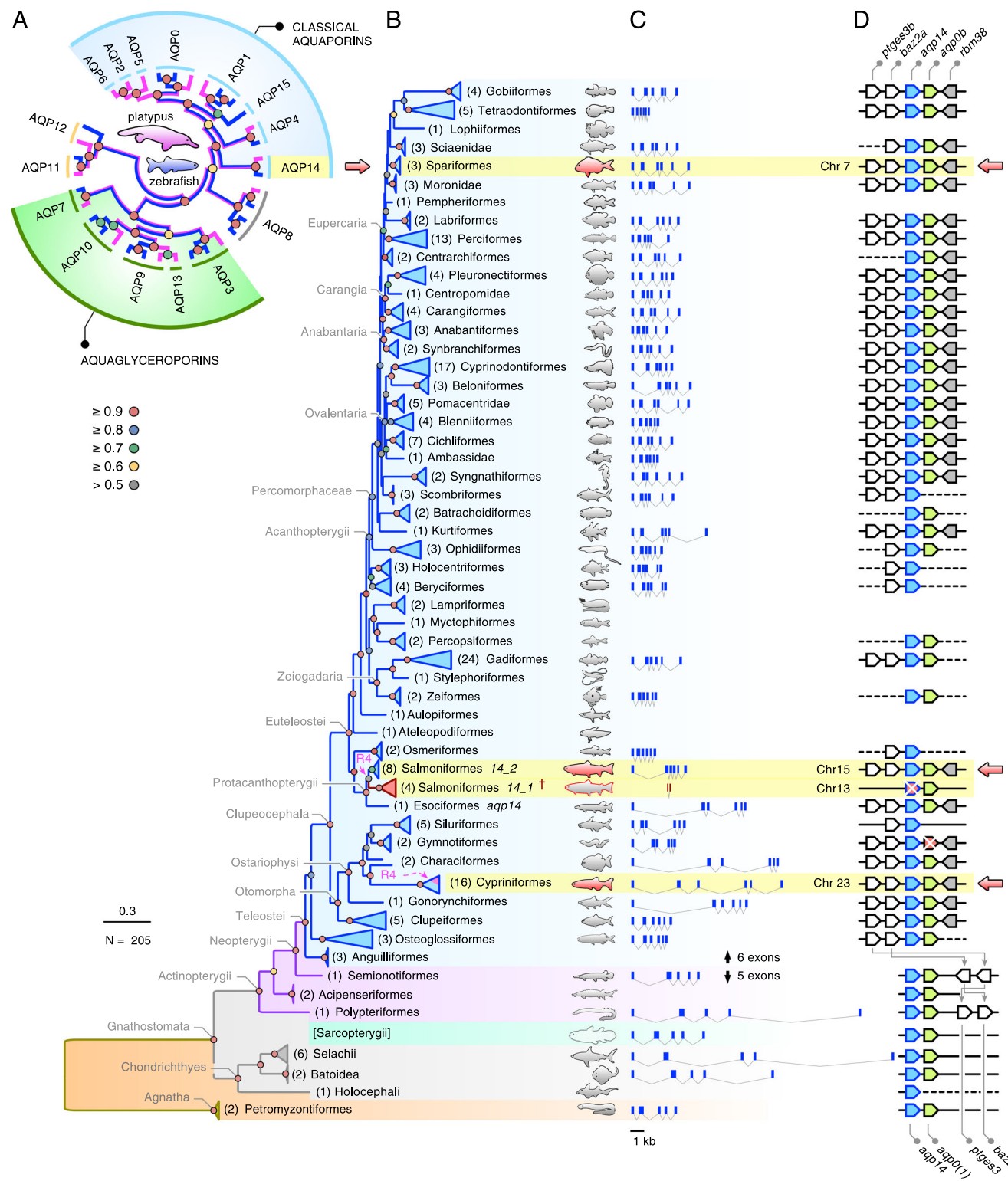

high-resolution tree with ~70% of posterior probabilities >0.9 (Fig. 1b; see Supplementary Fig. 1 for the fully annotated tree together with accession numbers). The tree topology almost completely recovers the phylogenetic interrelations of actinopterygian fishes recently determined from transcriptomic and genomic data[31]. This includes the position of anguilliforms (eels) relative to osteoglossiforms (bony tongues) as well as the clear hierarchical separation of the sister clade of Clupeocephala into Otomorpha and Euteleostei with respective subdivisions. The only major anomaly is the position of the Gobiiformes (mudskippers), which cluster with the Tetraodontiformes (box fishes) within the eupercarian clade. This is despite the mudskipper *aqp14* transcripts retaining lower identities (76.8 ± 0.2 %; $N = 4$) than those of other Percomorphaceae (80.5 ± 1.9%; $N = 87$) in relation to the tetraodontiform CDS.

Examination of hagfish (Hyperotreti) databases ($N = 2$) revealed only 1 encoded classical aquaporin (*aqp4*), and consequently the tree shows that the oldest forms of *aqp14* likely

**Fig. 1 Molecular phylogeny and structure of aquaporin genes. a** Mid-point rooted maximum likelihood tree (heuristic search optimised for parsimony; GTR model: NST = 6; partitioned by codon) of the major aquaporin subfamilies in Sarcopterygii represented by the platypus (magenta branches) and Actinopterygii represented by zebrafish (blue branches) illustrating the position of the *AQP14* subfamily next to *AQP4* in the classical grade of aquaporins. Since the *AQP10* gene is not currently found in the platypus genome, the metatherian Tasmanian devil *AQP10* ortholog is included. Bayesian inference (1,000,000 MCMC generations) yielded the same tree topology with posterior probability ranges indicated at each node. **b** Summarised Bayesian majority rule consensus tree (25 million MCMC generations inferred from 170,646 nucleotide sites) of *aqp14* channels from 59 orders of fishes. The tree is rooted with lamprey *aqp14*. R4 represents the fourth round of whole-genome duplication in salmonids and some cyprinids, with the salmonid *aqp14_1* pseudogenes[†] indicated in red. Key denotes the level of posterior probability at each node. Scale bar represents the expected substitutions per site. **c** Gene structures drawn to scale for representative species in each order. **d** Syntenic blocks for representative species in each order. Pointed ends of the gene symbol indicates coding direction, solid line indicates full linkage, dashed line indicates absence of scaffold data. Red gene symbols with a cross indicate pseudogenes. Arrows highlight lineages selected for functional experiments.

arose in a common ancestor of jawless and jawed vertebrates as reflected by petromyzontiform lampreys (Hyperoartia) and cartilaginous fishes (Chondrichthyes). The conserved *aqp14* subfamily (<40% amino acid differences between sharks and modern acanthomorph teleosts encompassing >500 million years of evolution since their last common ancestor[32]) has remained encoded in the genomes of all three major lineages of extant gnathostome vertebrates, the Chondrichthyes, Actinopterygii and Sarcopterygii. In the latter instance, this includes the Actinistia (coelacanths), Amphibia (caecilians, salamanders and frogs), Sauropsida (reptiles and birds), as well as prototherian and metatherian mammals (see below, Section on the Role of the C-terminus in the control of Aqp14 intracellular trafficking), but as noted previously, are fractionated into pseudogenes in eutherian mammals[16]. Further, despite clear evidence that teleost fishes retain many duplicated aquaporins resulting from a fish-specific whole-genome duplication (WGD)[16,26,29,30], only single-copy *aqp14* genes were discovered in the genomes of diploid species ($N = 182$), while intact duplicates and pseudogenes were detected in the genomes of paleotetraploid cyprinid ($N = 7$) and salmonid ($N = 4$) fishes, respectively.

The present evidence shows that the structures of the *aqp14* genes are split into five exons in basal lineages of actinopterygian, chondrichthyan and hyperoartian fishes as well as in all sarcopterygian animals that retain the genes, but into 6 exons in teleost fishes (Fig. 1c). Gene lengths are polymorphic, with the longest currently identified in the great white shark (*Carcharodon carcharias*, 20.74 kb), and the most compact in the Fugu (*Takifugu bimaculatus*, 1.35 kb), with the majority of *aqp14* gene structures in Euteleostei spanning <4.6 kb. All *aqp14* genes are located within a small, but conserved gene cassette upstream of each animal's *aqp0* paralog with rearrangement of the flanking *ptges3* and *baz2* paralogs to upstream loci in teleost fishes (Fig. 1d). While diploid teleost fishes are also known to retain two *aqp0* paralogs (*aqp0a* and −*0b*) and paleotetraploid teleosts such as Atlantic salmon four paralogs (*aqp0a1*, −*0a2*, −*0b1*, −*0b2*)[33], the *aqp14* genes are only found upstream of the *aqp0b* paralogs. This includes the *aqp14_1* pseudogenes upstream of the paleotetraploid salmonid *aqp0b1* paralogs and the complete *aqp14* gene upstream of a fractionated *aqp0b* pseudogene in the electric eel (*Electrophorus electricus*). These observations suggest loss of a WGD *aqp14(a)* gene shortly after the fish-specific WGD.

**Permeability properties of teleost Aqp14**. To investigate the functional properties of the *aqp14* genes, we isolated and cloned *aqp14* cDNAs from the freshwater ostariophysan zebrafish as a representative of the ancient lineage of Otomorpha, and two euteleostean species, the freshwater protacanthopterygian Atlantic salmon (*Salmo salar*) and the marine acanthopterygian gilthead seabream (*Sparus aurata*). Tertiary renders of the deduced proteins against a crystallographically resolved AQP4 structure mask (PDB: 3GD8), revealed that the Aqp14 channels retain the

canonical six transmembrane domains, together with two hemi-helices bearing NPA motifs, the 5 linking loops (3 external: A, C, E; 2 internal: B, D) and intracellular N- and C-termini (Fig. 2a). Some variation of the first NPA motif was noted amongst gadi-form, ophidiiform and kurtiform teleosts, which encode NPS in that position. In contrast to other classical aquaporins, which retain four ar/R residues, the Aqp14 renders indicate that a fifth residue (Val[29] in seabream, or Val[30] in zebrafish and salmon) on TMD1 participates in the outer vestibule constriction of this subfamily (Fig. 2a). In addition, although the Arg constriction residue following the second NPA motif is fully conserved, none of the other Aqp14 constriction residues are aromatic.

The biophysical properties of the zebrafish, salmon and seabream channels were investigated using the *Xenopus laevis* oocyte-swelling and radioactive or fluorescent substrate uptake assays. Oocytes expressing either the zebrafish, seabream or salmon channels (DrAqp14, SaAqp14 or SsAqp14, respectively) show an approximately 5-fold increase in osmotic water permeability ($P_f$) with respect to the water-injected oocytes, as well as a conserved inhibitiory sensitivity to mercury, which is reversed by the reducing agent β-mercaptoethanol (Fig. 2b). Oocytes injected with SaAqp14 and SsAqp14 are also slightly permeable to glycerol, urea, hydrogen peroxide and the ammonia analogue methylamine (Fig. 2c–f). By contrast, oocytes expressing DrAqp14 are highly permeable to urea and hydrogen peroxide, but as for SaAqp14 and SsAqp14 show a lesser permeability to glycerol and methylamine (Fig. 2c–f). These data show that in contrast to other classical aquaporins, Aqp14 channels share some of the permeation properties typically elicited by aquaglyceroporins.

**Tissue distribution and cellular localization**. As a first insight to the physiological function of the Aqp14 channels, we investigated the tissue distribution and cellular localization in the three selected species. Expression of *aqp14* transcripts was investigated by reverse-transcriptase PCR (RT-PCR) using paralog-specific primers on extracts from the brain, eye, lens, ovary, testis, gills, kidney, liver, anterior intestine, mid intestine, rectum and skin (Fig. 3a–c; and Supplementary Fig. 2 for biological replicates). In each species, prominent *aqp14* expression is detected in the brain, lens and testis, but in other tissues it shows greater variability between the species, and in some cases also between animals. For example, although *aqp14* mRNA is found in the gills, kidney, liver, gut and skin of zebrafish and salmon, the expression of *aqp14* in these tissues from seabream is less prominent or undetectable in some animals (Fig. 3a–c).

To determine the expression and cellular localization of Aqp14 proteins in the different tissues we produced an affinity-purified rabbit antibody against a C-terminal 16 amino acid region of the seabream channel (Supplementary Fig. 3a–c). The specificity of the antiserum was confirmed through western blot analysis of total membrane protein extracts from oocytes injected with equal

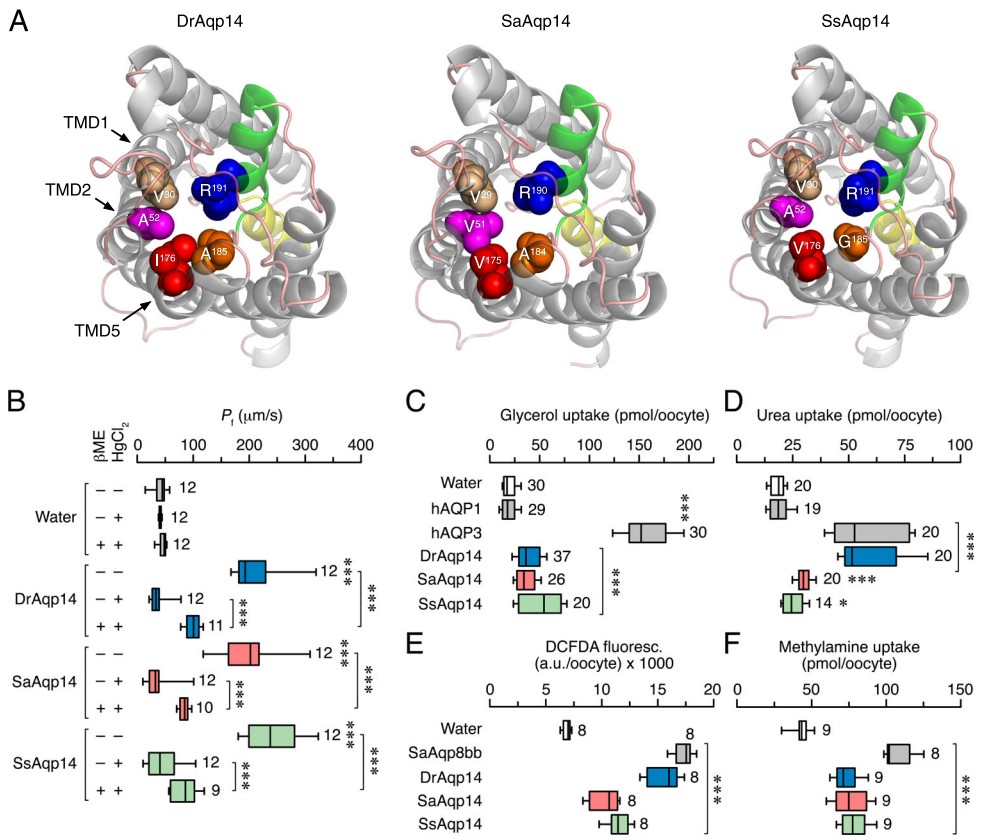

**Fig. 2 Structure and function of piscine Aqp14 channels. a** Extracellular views (cartoon renders) of zebrafish (DrAqp14), gilthead seabream (SaAqp14) and Atlantic salmon (SsAqp14) channels illustrating the quintet of Arg (R) and non-aromatic constriction residues (spacefill) on transmembrane domains (TMD) −1, −2, −5 and loop E. TMDs are grey, and hemihelices bearing the first and second NPA motifs are yellow and green, respectively. **b–f** Osmotic water permeability ($P_f$; B) and substrate uptake (C, [$^{3}$H] glycerol; D, [$^{14}$C] urea; E, $H_2O_2$ using the CM-$H_2$DCFDA reagent; and F, [$^{14}$C] methylamine) of *X. laevis* oocytes injected with water (control) or 15 ng of cRNA encoding DrAqp14, SaAqp14 or SsAqp14. In B, the $P_f$ was assayed in the presence or absence of $HgCl_2$ and β-mercaptoethanol (βME). In C-D, oocytes injected with human AQP1 (hAQP1) or −3 (hAQP3) were used as negative or positive controls, respectively, whereas in **e** and **f**, oocytes expressing seabream Aqp8bb were the positive controls. Data are box and whisker plots with the number of biologically independent oocytes indicated beside each plot. *$P < 0.05$; **$P < 0.01$; ***$P < 0.001$, statistically different (unpaired Student's *t*-test), with respect to control oocytes, or as indicated in brackets.

amounts of seabream *aqp0a*, −*1aa*, −*1ab*, −*4a*, −*8bb*, −*3a*, −*9b*,. −*7*, −*10b* and −*14* cRNA (Supplementary Fig. 3d). The cross-reactivity of the antiserum against DrAqp14 and SsAqp14, which, respectively, show 63 and 81% identity in the region of the SaAqp14 C-terminus selected for immunization, was also tested by Western blot analysis on total membrane protein extracts from *X. laevis* oocytes injected with equal amounts of *aqp14* cRNA. The results show that the antiserum generated was able to specifically recognize Aqp14 from all three species, although with less intensity to that of SaAqp14 according with their lower level of amino acid identity within the immunization region (Supplementary Fig. 3e).

Immunoblotting analyses of zebrafish tissues using the seabream Aqp14 antibody revealed a single immunoreactive polypeptide band of approximately the same molecular mass as the predicted Aqp14 monomer (29.5 kDa) in the brain, eye, lens, testis, rectum and gills, somewhat more intense when normalized to prohibitin expression in the brain, eye and gills (Fig. 3d, Supplementary Fig. 4d–f). In salmon, two immunoreactive bands of approximately 29 and 33 kDa were detected in the same tissues, which likely correspond to the predicted monomer (29.4 kDa) and post-translational modifications of this product, respectively (Fig. 3e). In the lens, however, only a single immunoreactive band with an intermediate molecular mass of approximately 31 kDa was detected. In salmon, the amount of the

29 or 33 kDa polypeptides in the brain, eye and gills, and of the 31 kDa band in the lens, seemed to be slighty higher than in the other tissues as observed in the zebrafish. Finally, in the seabream a strong 29 kDa band corresponding to the predicted monomer (29.3 kDa) was observed in all tissues with comparable intensity, except in the gills, where this band was much less intense (Fig. 3f). In all tissues, except in the lens, additional bands of ~50 kDa or higher, weakly reacting with the Aqp14 antibody, were also noticed, which could correspond to dimerization products and/or complex post-translational modifications. For the three species, the specificity of the reactions was confirmed by the preadsorb-tion of the Aqp14 antiserum with the immunizing peptide (Fig. 3e–g).

The cellular localization of Aqp14 in the different tissues of the three species was determined by immunofluorescence micro-scopy. These experiments revealed that the cell-type expression of Aqp14 in the brain, lens, retina, gills, rectum, testis and muscle of the three teleost species is highly conserved (Fig. 4; see also Supplementary Figs. 5–8). In the brain, Aqp14 is distributed exclusively in the optical tectum in the stratum marginale and stratum periventriculare (Fig. 4a,b), whereas in the eye the channel is detected in the lens epithelium but not in the lens fiber cells (Fig. 4c). The Aqp14 channel is also expressed in ganglion cells located in the inner surface (the ganglion cell layer) and in neurons of the inner nuclear layer of the retina, which extend

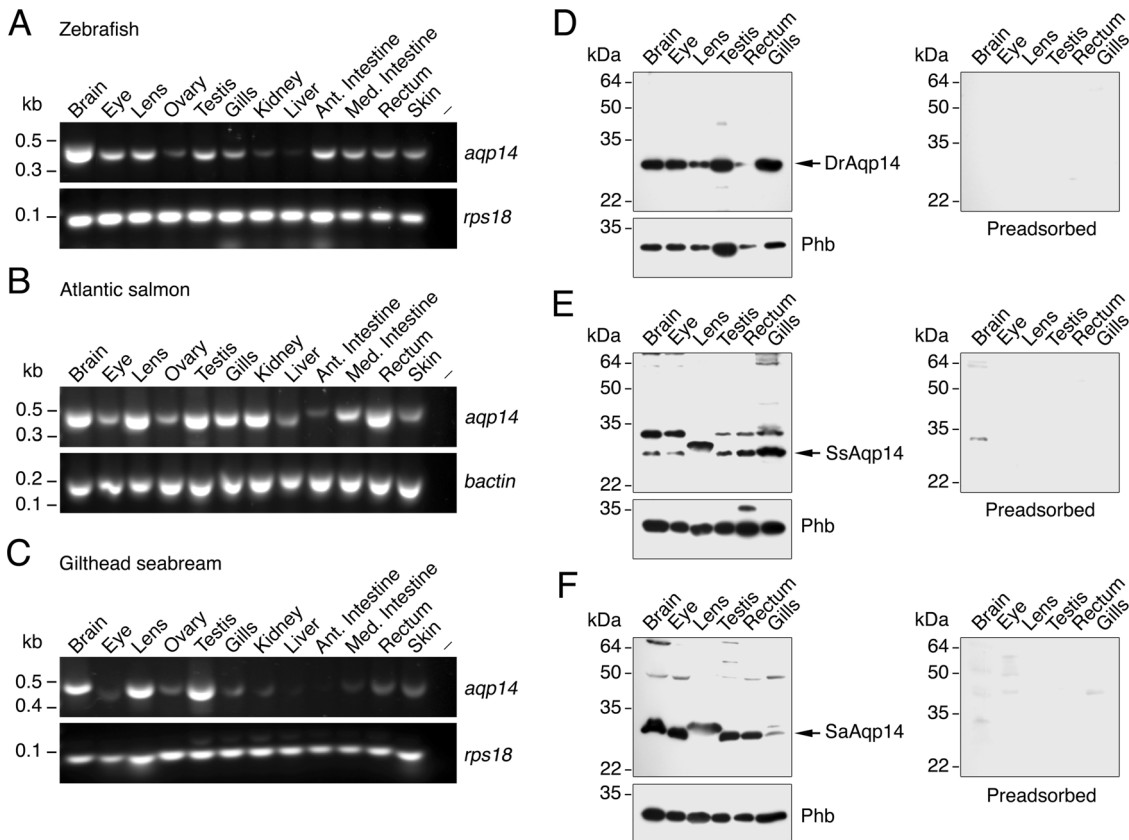

**Fig. 3 Expression of Aqp14 in different teleost tissues. a–c** Representative RT-PCR analysis of *aqp14* gene expression in different tissues from zebrafish (**a**), Atlantic salmon (**b**) and gilthead seabream (**c**) ($N = 3$ fish as indicated). The minus indicates absence of RT during cDNA synthesis. The size (kb) of PCR products and molecular markers are indicated on the left. Uncropped gels are shown in Supplementary Fig. 2. **d–f** Representative immunoblots of Aqp14, and prohibitin (Phb) as loading control, in protein extracts from brain, eye, lens, testis, rectum and gills from zebrafish (**d**), Atlantic salmon (**e**) and gilthead seabream (**f**) using the SaAqp14 antibody. In each blot, arrows indicate aquaporin monomers based on in silico determination of molecular masses of DrAqp14 (29.5 kDa), SsAqp14 (29.4 kDa) and SaAqp14 (29.3 kDa). Duplicated blots were run in parallel and incubated with the primary antibody preadsorbed by the antigenic peptide to test for specificity. Molecular mass markers (kDa) are on the left. Uncropped blots are shown in Supplementary Fig. 3d, e.

their axons across the inner plexiform layer (Fig. 4d). In the gills, Aqp14 immunostaining is found in the apical plasma membrane of epithelial cells along the length of gill lamellae (Fig. 4e). In the testis, Aqp14 expression is found exclusively in the myofibroblasts and smooth muscle cells of the tunica albuginea (Fig. 4f and inset), whereas the channel is also expressed in the myocytes of white muscle fibers (Fig. 4g). Interestingly, in both the zebrafish and salmon that were maintained in freshwater, very low levels of Aqp14 are detected in rectal enterocytes, while prominent Aqp14 expression is observed both in the cytoplasm and apical plasma membrane of these cells in seawater-held seabream (Fig. 5a–c). In the seabream rectum, the channel is also detected in some cells of the submucosa, but not in the mucus secreting goblet cells spread in the luminal epithelium (Fig. 5c). The sections of all these tissues incubated with the preabsorbed affinity-purified Aqp14 antiserum did not show any positive signals (Figs. 4h–n and 5d–f), confirming the specificity of the antibody. In contrast, in the three species investigated no consistent Aqp14 immunostaining is observed in the ovary, kidney, liver, anterior and mid intestine, and skin.

**C-terminal control of Aqp14 intracellular trafficking.** The observation that Aqp14 in a marine teleost, such as the seabream, is strongly expressed in both the cytoplasm and apical plasma membrane of rectal enterocytes, which is a major site of intestinal

water resorption in marine teleosts, prompted us to investigate the potential role of the Aqp14 cytoplasmic C-terminus in the control of the channel's intracellular trafficking. Amino acid sequence alignment of zebrafish, salmon and seabream Aqp14 revealed the presence of two conserved Thr residues (Thr$^{240}$/Thr$^{241}$ and Thr$^{259}$/Thr$^{260}$) and one Ser (Ser$^{256}$/Ser$^{257}$) in the C-terminus of the channels as potential sites for protein kinase C (PKC) and protein kinase A (PKA) phosphorylation, respectively (Fig. 6a). To experimentally assess if PKC and PKA regulate the trafficking of Aqp14, *X. laevis* oocytes expressing DrAqp14, SaAqp14 or SsAqp14 were exposed to the PKC activator phorbol 12-myristate 13-acetate (PMA) or the cAMP-PKA activator forskolin (FSK), the latter after preincubation of oocytes with the phosphodiesterase inhibitor 3-isobutyl-1-methylxanthine (IBMX). Interestingly, the results of these experiments showed that the $P_f$ of Aqp14 oocytes is inhibited by ~45% by PMA regardless of the ortholog expressed, whereas treatment with IBMX and FSK stimulates water permeability by ~40–60% (Fig. 6b). Immunoblots of total and plasma membrane extracts of these oocytes revealed that the opposite effects of PMA and IBMX/FSK on oocyte permeability is caused by the PKC-mediated negative and PKA-mediated positive regulation of Aqp14 trafficking to the oocyte surface (Fig. 6c, Supplementary Fig. 9).

To corroborate that the effects of PKC and PKA on Aqp14 trafficking occur via the Thr and Ser residues identified in the

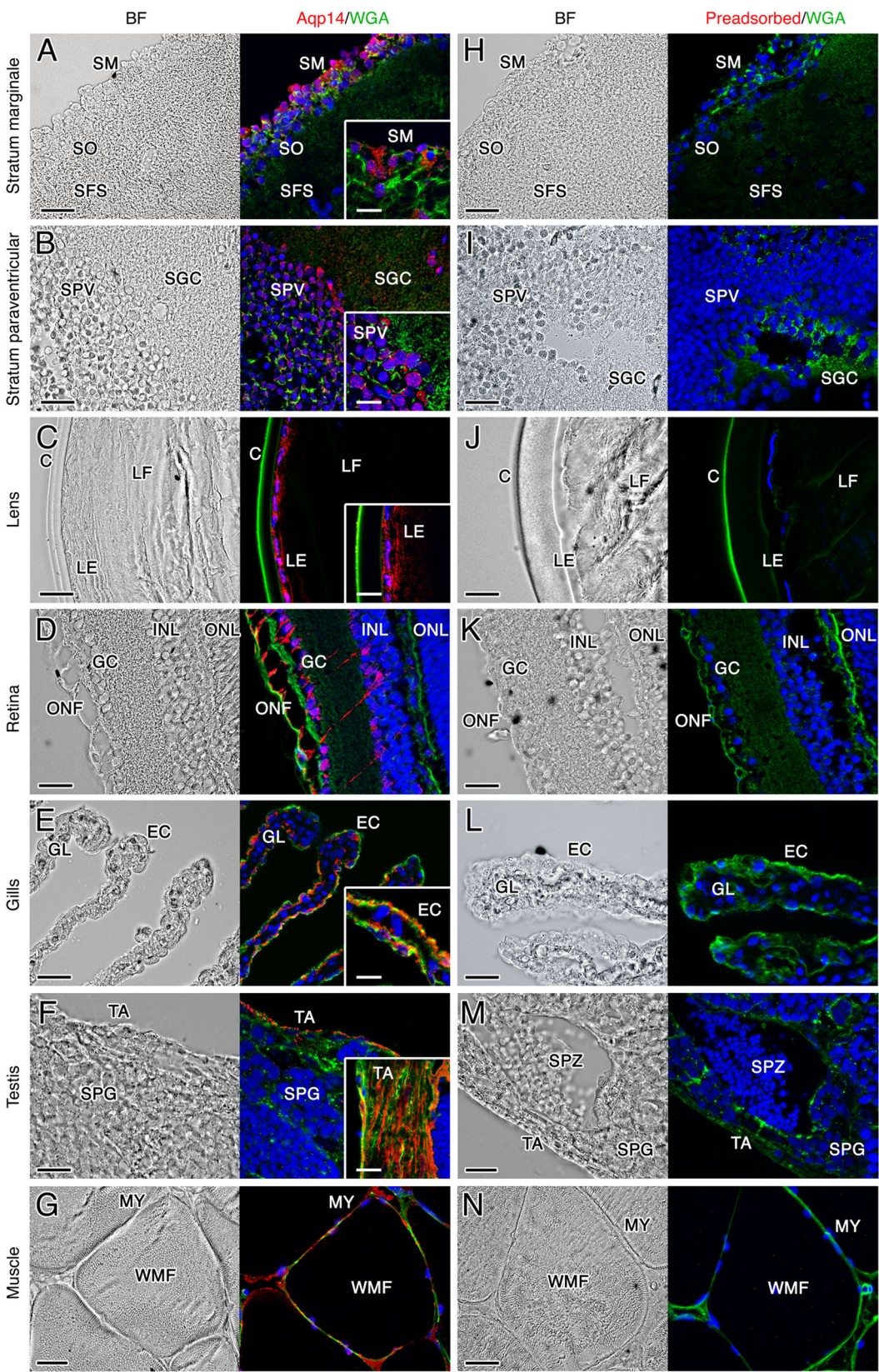

C-terminus of the piscine channels, oocytes were injected with wild-type DrAqp14, or with DrAqp14 independently mutated into Ala or Asp at $Thr^{241}$, $Ser^{257}$ and $Thr^{260}$ residues to, respectively, mimic nonphosphorylated and phosphorylated states, and treated with PMA or IBMX/FSK as above. Swelling assays showed that DrAqp14-T241A and -S257D constitutively

increase water permeability with respect to the wild-type by ~50% and ~90%, respectively (Fig. 6d, Supplementary Fig. 10e). In addition, these mutants respectively prevent the effect of PMA or FSK, whereas the DrAqp14-T260A and -T260D mutants have no effect. According to these observations, the DrAqp14-T241D mutant reduces the permeability with respect to the wild-type by

**Fig. 4 Immunolocalization of Aqp14 in different tissues from zebrafish, Atlantic salmon and gilthead seabream. a–g** Representative brightfield and immunofluorescence microscopy images of Aqp14 in the seabream and zebrafish optical tectum of the brain (**a** and **b**, respectively), zebrafish lens (**c**) and retina (**d**), seabream gills (**e**) and testis (**f**), and Atlantic salmon muscle (**g**). Sections were labeled with affinity-purified seabream Aqp14 antiserum (red) and counterstained with 4′,6-diamidino-2-phenylindole (DAPI; blue) and wheat germ agglutinin (WGA) (green). **h–n** Control sections incubated with preabsorbed antiserum were negative. Scale bars: 20 μm (10 μm insets). SM stratum marginale, SO stratum opticum, SFS stratum fibrosum superficiale, SPV stratum periventriculare, SGC stratum griseum centrale, C capsule, LE lens epithelium, LF lens fiber, ONF optic nerve fiber, GC ganglion cells, INL, inner nuclear layer, ONL outer nuclear layer, GA gill arch, EC epithelial cell, GL gill lamellae, TA tunica albugínia, SPG spermatogonia, SPZ spermatozoa, WMF white muscle fiber, MY myocyte.

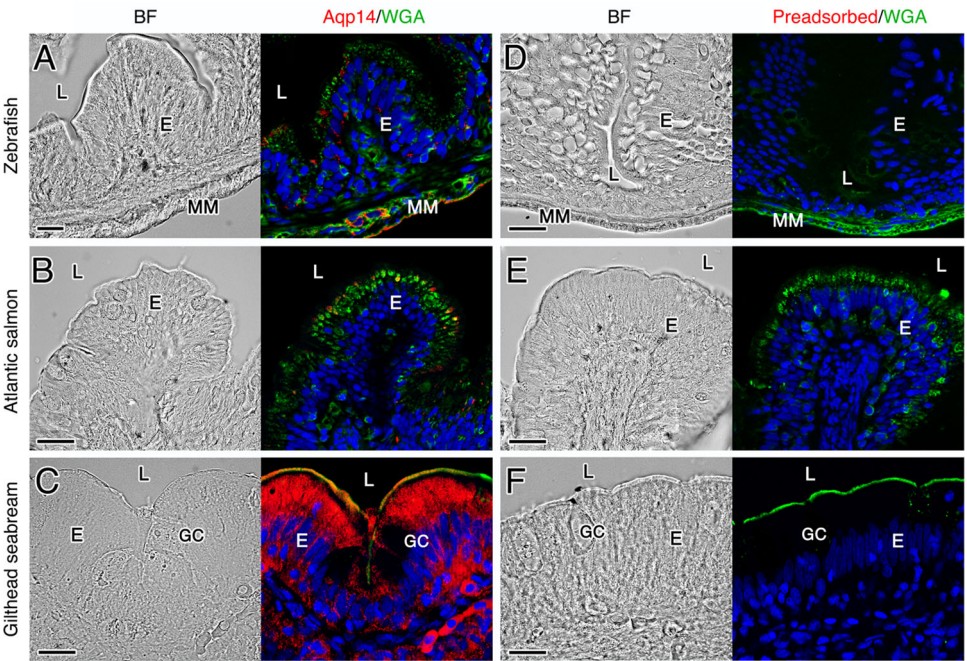

**Fig. 5 Immunolocalization of Aqp14 in zebrafish, Atlantic salmon and gilthead seabream rectum. a–c** Representative brightfield and immunofluorescence microscopy images of Aqp14. Sections were labeled with affinity-purified seabream Aqp14 antiserum (red) and counterstained with 4′,6-diamidino-2-phenylindole (DAPI; blue) and wheat germ agglutinin (WGA) (green). **d–f** Control sections incubated with preabsorbed antiserum were negative. Scale bars: 20 μm. L lumen, E enterocyte, GC Goblet cell, MM muscularis mucosa.

~53%, and blocks the inhibition by PMA, while it does not interfere with the positive effect of FSK. By contrast, the DrAqp14-S257A mutant does not change the negative effect of PMA on channel trafficking, but abolishes the positive effect of FSK. Immunoblot analysis confirmed that the wild-type and mutant channels were expressed in oocytes at similar levels (Fig. 6e). The same results were obtained when oocytes expressed wild-type SaAqp14 or the SaAqp14-T240A, -T240D, -S256A and S256D mutants (Fig. 6f, g, Supplementary Fig. 10g). These data therefore suggest that the intracellular trafficking of teleost Aqp14 is dually regulated by PKC and PKA pathways through phosphorylation of $Thr^{240}/Thr^{241}$ and $Ser^{256}/Ser^{257}$, respectively. Strikingly, these residues are not only fully conserved in the C-termini of the Aqp14 orthologs of all agnathan and gnathostome fishes, but also in nearly all of the sarcopterygian orthologs (see Supplementary Fig. 11). Available genomic data thus reveal that all extant lineages of tetrapods, except eutherian mammals, also encode the *AQP14* channels upstream of the *AQP0* gene, but indicate that some birds and the platypus have substituted the $Thr^{240}/Thr^{241}$ for an Ala. These observations suggest strong purifying selection of the PKC and PKA targeted phosphorylation sites in Aqp14 for >500 million years of evolution.

**Vasotocinergic and isotocinergic regulation of Aqp14.** Since seawater acclimation in euryhaline teleosts is associated with the increased expression of some aquaporin orthologs in the intestinal and rectal epithelial cells[34], the Aqp14 immunostaining pattern in seabream rectum raised the question of a potential regulation of Aqp14 trafficking by environmental salinity. In teleosts, osmoregulatory adaptations are regulated by the pituitary neurohormones vasotocin (AVT) and isotocin (IT), which evolutionarily, are amongst the most primitive molecules involved in vertebrate osmoregulation[35]. AVT and IT are orthologs of mammalian arginine vasopressin and oxytocin, respectively, which act on target tissues through specific AVT1a and AVT2 receptors (AVT1aR and AVT2R), preferentially coupled to PKC- or PKA-mediated pathways, respectively, or the IT receptor (ITR) that also acts via PKC[36–38]. We therefore investigated the possible control of seabream Aqp14 intracellular transport by AVT and IT, and whether this regulation involves the same C-terminal Thr and Ser residues identified above.

To examine this hypothesis, *X. laevis* oocytes were co-injected with seabream wild-type Aqp14, Aqp14-T240A or -S256A and seabream AVT1a2R, AVT2R or ITR. The cDNAs of the latter neuropetide receptors were previously isolated from publically

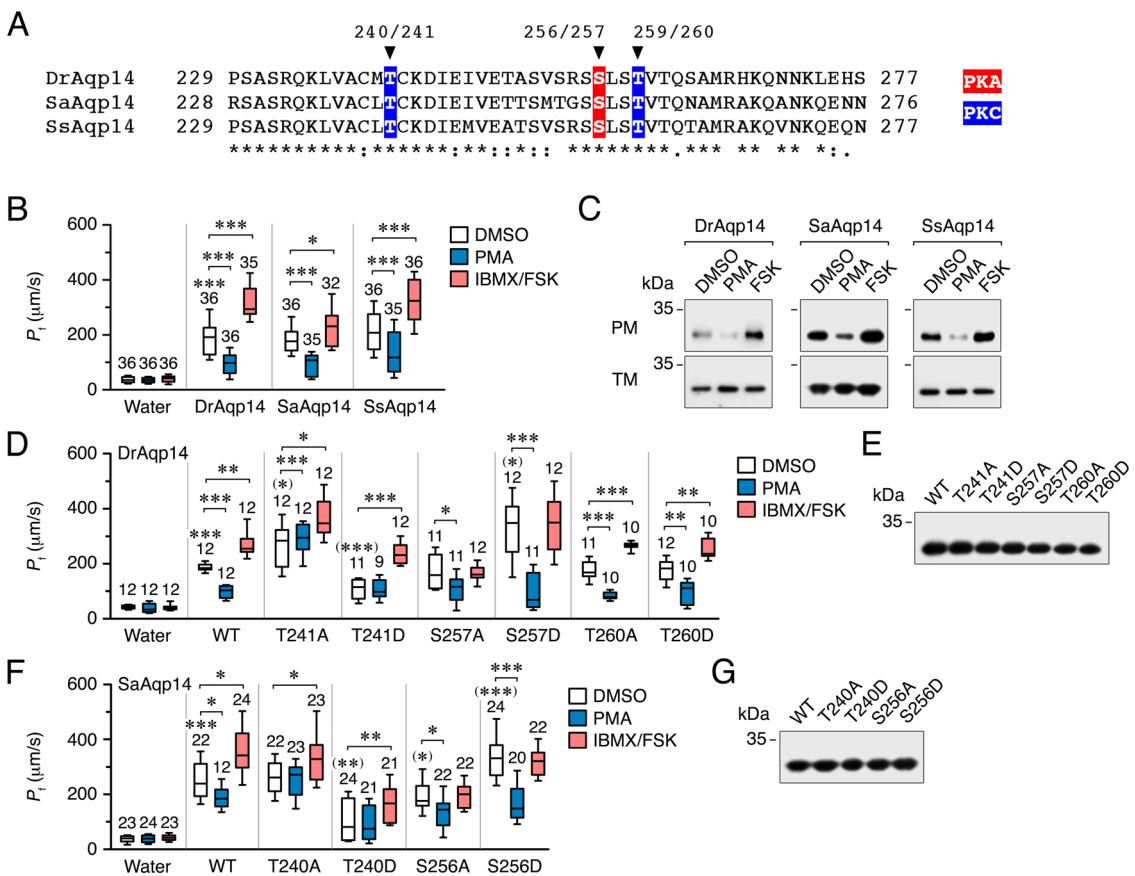

**Fig. 6 Regulation of Aqp14 intracellular trafficking in *X. laevis* oocytes. a** Amino acid alignment of the C-terminus of Aqp14 from zebrafish (DrAqp14), seabream (SaAqp14) or Atlantic salmon (SsAqp14). Putative phosphorylation sites by PKC (highlighted in blue) or PKA (highlighted in red) are indicated. **b** Osmotic water permeability ($P_f$) of oocytes injected with water (control) or DrAqp14, SaAqp14 or SsAqp14 cRNAs, and exposed to the PKC activator PMA (100 nM) or the PKA activator FSK (100 μM), the latter after 1 h incubation with the phosphodiesterase inhibitor IBMX (100 μM), or to the drug vehicle (DMSO, control). **c** Representative immunoblot of total and plasma membrane (TM and PM, respectively) protein extracts from oocytes treated as in **b** using the seabream Aqp14 antiserum. **d-f** $P_f$ of oocytes injected with water or expressing wild-type DrAqp14 or SaAqp14 (WT), or mutant DrAqp14 or SaAqp14 at the putative PKC and PKA phosphorylation residues, and treated with PMA or IBMX/FSK as in **b**. **e-g** Representative immunoblot of TM protein extracts from oocytes injected with each construct showing equivalent expression. In **b**, **d** and **f**, data box and whisker plots with the number of biologically independent oocytes indicated above each plot. *$P < 0.05$; **$P < 0.01$; ***$P < 0.001$, statistically different (unpaired Student's $t$-test), with respect to control oocytes, with respect to the WT (in parenthesis), or as indicated in brackets. Uncropped blots in **c**, **e** and **g** are shown in Supplementary Figs. 9 and 10.

available data[39]. Oocytes were then exposed to AVT or IT in the presence or absence of the PKC inhibitor Bisindolylmaleimide (Bim-II) or the PKA inhibitor H89. The results show that the $P_f$ of oocytes expressing the wild-type Aqp14 plus AVT1a2R or ITR are inhibited by ~84% and ~65% by AVT and IT, respectively, regardless of the presence of H89, whereas both effects are abolished when oocytes are exposed to Bim-II, or they co-express the AVT1a2R or ITR with the Aqp14-T240A mutant (Fig. 7a, c). In contrast, water permeability of oocytes expressing wild-type Aqp14 and AVT2R is increased by ~70% after AVT treatment, while this effect is prevented with H89, but not with Bim-II, or by coexpressing the Aqp14-S256A mutant instead of the wild-type (Fig. 7b). Oocytes expressing the Aqp14-T240A or -S256A mutants alone show more and less $P_f$, respectively, than those injected with the wild-type, suggesting some level of endogenous activation of PKC or PKA in the oocytes (Fig. 7a–c). These data are therefore consistent with previous mutagenesis experiments, and suggest that activation of the AVT2R can induce the insertion of Aqp14 into the plasma membrane through PKA-mediated Ser[256] phosphorylation, whereas the AVT1a2R and ITR inhibit channel trafficking to the oocyte surface through PKC-mediated Thr[240] phosphorylation.

## Discussion
In the present contribution, we uncover the broad genomic diversity and function of a novel subfamily of vertebrate water channel genes that currently lack formal annotation in genome databases. In accordance with the chronology of aquaporin gene discovery, the subfamily is named *AQP14*, after *AQP13*, which was originally isolated from frog oocyes[15], and shown to exist in amphibians and prototherian mammals[16]. Our previous phylogenetic analyses revealed that *aqp14* genes are members of the classical Aqp4-related grade of aquaporins, with multiple aqp4-like genes traced to cnidarian lineages of jellyfishes and corals[16]. The classical AQP4-related grade of aquaporins, thus evolved prior to the separation of Deuterostomia and Protostomia, and is distinct from the AQP8-related grade of channels, which also evolved prior to the separation of these lineages[16,19,20,40]. Our new data show that the *aqp14* genes are located upstream of the *aqp0* paralog in nearly all vertebrate lineages, except hagfishes. The current evidence that hagfishes lack the *aqp14* channel is consistent with the absence of the genomic region based upon synteny to other vertebrate genomes, and the absence of an *aqp0* paralog in these jawless craniates. The observed paucity of classical aquaporins in hagfish may reflect their reduced

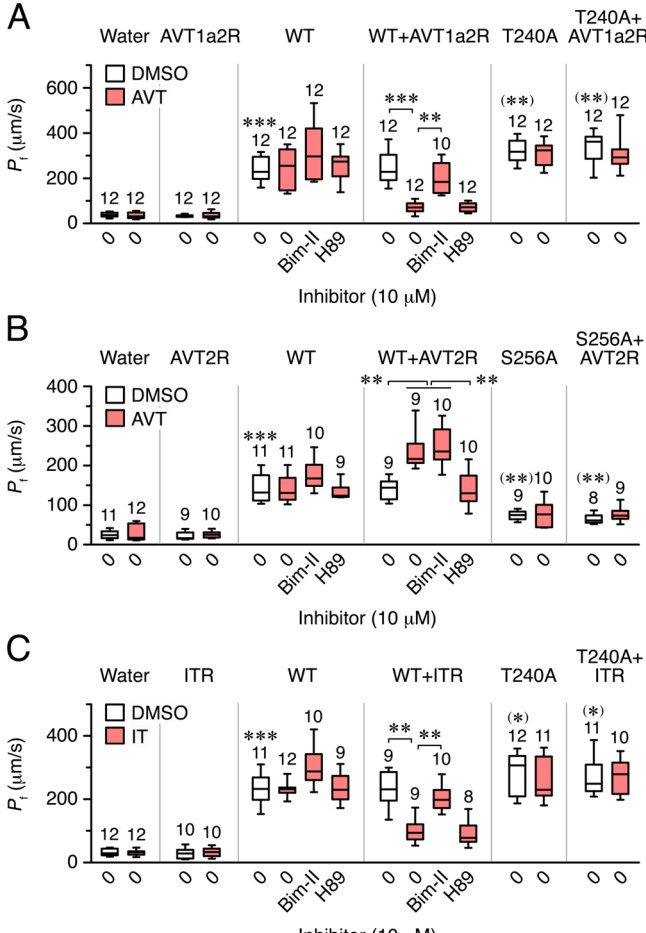

**Fig. 7 Vasotocin (AVT) and isotocin (IT) regulation of seabream Aqp14 in *X. laevis* oocytes.** Osmotic water permeability ($P_f$) of oocytes expressing or not the seabream AVT1a2R (**a**), AVT2R (**b**) or ITR (**c**) receptors, and injected with water, wild-type Aqp14 (WT), or the Aqp14-T240A or -S256A mutants. Oocytes were treated with DMSO (controls) or 10 μM AVT or IT, in the presence or absence of 10 μM of PKC and PKA inhibitors (Bim-II and H89, respectively). Data are box and whisker plots with the number of biologically independent oocytes indicated above each plot. *$P < 0.05$; **$P < 0.01$; ***$P < 0.001$ (unpaired Student's t-test), with respect to control oocytes, with respect to the WT (in parenthesis), or as indicated in brackets.

osmoregulatory demands, since unlike other vertebrates, hagfishes are marine osmoconformers with blood osmolalities isosmotic to seawater[41]. By contrast, extant lampreys are osmoregulators in both freshwater and marine environments, and encode the *aqp14* gene upstream of their fused *aqp01* paralog[16]. Indeed as in marine teleosts, lampreys that live in marine environments are hyposmotic to seawater and drink to compensate for water loss revealing that such an osmoregulatory adaptation is likely ancient[42].

It has long been established that a neuropeptide-regulated water channel response governs antidiuresis or transepithelial water uptake in tetrapods, whereby vasopressin/vasotocin and the V2-type receptor induces cAMP-mediated trafficking of AQP2 in the kidney collecting tubules of mammals, birds and frogs, and of AQP5 and −6 channels in the skin of frogs[27,43–46]. However, these aquaporin gene clusters are unique to sarcopterygian animals[16], and consequently, the broad prevalence and conservation of the *aqp14* gene in vertebrate genomes suggests that this subfamily of water channels represents a much older system that

evolved under strong selective pressure to mediate osmoregulatory mechanisms. We identify two residues in the C-terminus of Aqp14 that provide a clue to this trait. We show that a Thr induces the retention of Aqp14 through AVT-AVT1a2R and IT-ITR mediated activation of the PKC pathway, while a Ser alternatively induces membrane insertion of Aqp14 through AVT-AVT2R-mediated activation of the PKA pathway. In addition, our data reveal that the Aqp14 channel is highly expressed in the cytoplasm and apical plasma membranes of rectal enterocytes of the marine teleost, gilthead seabream, a region of the intestine that plays a major role in the resorption of water in seawater adapted teleosts[34,47]. The high conservation of the Thr and Ser residues in both piscine and tetrapod orthologs, and their respective regulation by the PKC and PKA transduction pathways in distantly related fishes, indicates that the Aqp14 channels co-evolved with vasotocinergic and isotocinergic neuropeptide systems to regulate water homeostasis throughout vertebrate evolution.

In contrast to other classical aquaporins, our data show that Aqp14 is a polytransporter, with permeabilites to urea, ammonia, $H_2O_2$ and glycerol in addition to water. This is unusual for classical aquaporins, which primarily transport water while preventing proton flux due to the specifc arrangement of the ar/R and NPA constriction residues[48]. Molecular renders of the piscine Aqp14 channels revealed that they each present a quintet of small non-aromatic constriction residues with hydrophobic side chains. This provides a more open and less polar arrangement compared to AQP1 and AQP4 channels, which in vertebrates only present four ar/R constriction residues that are highly selective for water due to the conserved His on TMD5[40,49,50]. The arrangement of the five constriction residues in the Aqp14 channels also differs from other aquaporins that present ar/R quintets, including bacterial AqpZ, plant TIP2;1 and vertebrate AQP8[51]. In these latter channels, an interacting fifth residue on loop C contributes to the selectivity filter[51], while in Aqp14, the fifth residue is located on TMD1. This novel topology generates the hydrophobic pockets in the Aqp14 channels to functionally resemble the selectivity filters of aquaglyceroporins[49], and although transport efficiencies differ between species, it likely explains their broader permeability properties for urea and glycerol. Since the Aqp14 channels are expressed in the cell membranes of neurons, muscles and gill epithelia, this feature indicates that the channels may also play important roles in metabolite and nitrogen metabolism.

In conclusion, we uncover the genomic diversity, structure and function of a novel subfamily of water channels in vertebrates. The *aqp14* gene is located upstream of the *aqp0* paralog in all extant vertebrates, except agnathan hagfishes and eutherian mammals, which in some of the latter lineages retain *AQP14* pseudogenes. Our phylogenetic data show that the *aqp14* subfamily clusters with the *aqp4*-related grade of classical aquaporins, but in contrast to other classical aquaporins, the proteins form hydrophobic selectivity filters through the presentation of five constriction residues on TMD1, −2, −5 and loop E. This unique arrangement supports our experimental data demonstrating that the channel is a polytransporter facilitating the molecular permeation of water, urea, ammonia, $H_2O_2$ and glycerol. We further show that channel trafficking is tightly controlled by the ancient vasotocinergic and isotocinergic neuropeptide systems to activate PKC and PKA transduction pathways for the phosphorylation of highly conserved Thr and Ser residues in the C-terminus. These data coupled with observations that the Aqp14 channel is highly expressed in rectal enterocytes of a marine fish, which is the major site of water absorption in seawater adapted fishes, reveal that the neuropeptide regulation of the Aqp14 channels predates the vasotocinergic/vasopressin regulation of AQP2, AQP5 or AQP6 observed in tetrapods and

thus represents an ancestral mechanism of neuropeptide-regulated water homeostasis in vertebrates.

## Methods

**Fish and chemicals**. Adult ~1 kg gilthead seabream (*Sparus aurata*) raised in captivity were maintained in seawater (~3.0% salinity) at the facilities of the Institute of Marine Sciences, Spanish Council for Scientific Research (CSIC, Spain) following previously described procedures[52,53]. Adult zebrafish were obtained from the PRBB Animal Facility (Barcelona, Spain), whereas ~1.5 kg Atlantic salmon were collected from the Matredal Aquaculture Research Station (61 °N) in Norway. Both species were maintained in freshwater (<0.05% salinity). For tissue collection, fish were sedated with 500 ppm of phenoxyethanol or 10 mg/l metomidate (Syndel, Victoria, BC, Canada), weighted and immediately euthanized by decapitation. Biopsies of different tissues were either frozen in liquid nitrogen and stored at −80 °C for RNA and protein extraction, or processed for histology and immunofluorescence microscopy (see below). Procedures relating to the care and use of animals and sample collection were carried out in accordance with the protocols approved by the Ethics Committee (EC) of the Institut de Recerca i Tecnologia Agroalimentàries (IRTA, Spain) following the European Union Council Guidelines (86/609/EU), or in accordance with the regulations approved by the governmental Norwegian Animal Research Authority (http://www.fdu.no/fdu/).

All chemicals were purchased from Sigma–Aldrich (St Louis, MO, USA) unless indicated otherwise.

**cDNA cloning**. Gilthead seabream and Atlantic salmon *aqp14* genes were identified by screening the corresponding genomes[54,55], and gene specific oligonucleotide primers flanking the first and last exon were designed to isolate the full-length mRNA coding sequences by reverse transcriptase-(RT)-PCR. For zebrafish *aqp14*, 5′ and 3′end specific primers were designed using a publically available sequence (GenBank accession no. XM_005174125). The sequences of the primers contained EcoRV and SpeI restriction sites were as follows: for seabream, forward and reverse primers were 5′-gatatcGGCTGCTCTCATCAAAGGAC-3′ and 5′-actagtCAGTCAAGCCTGATCTACACGA-3′, respectively; for Atlantic salmon, forward and reverse primers were 5′-gatatcCATGGCGATTCGAGAGGAGT-3′ and 5′-actagtACTAGTGTTGTGTAGCATGA-3′, respectively; and for zebrafish, forward and reverse primers were 5′-gatatcGCATAGACCAACTGACGGATT-3′ and 5′-actagtTGTGTGACTGACTACATGTGCAA-3′, respectively. Total RNA from brain (seabream and zebrafish) or kidney (Atlantic salmon) was purified using the GenElute mammalian total RNA miniprep kit (Sigma–Aldrich), according to the manufacturer's instructions. cDNA synthesis was performed with 1 μg of total RNA using an oligo dT(12–18) primer (Life Technologies Corp., Carlsbad, CA, USA) and SuperScript II RT enzyme (Life Technologies Corp.) as previously described[52]. The PCR was carried out with the EasyA™ high-fidelity PCR cloning enzyme (Agilent Technologies, Santa Clara, CA, USA), with an initial denaturing step for 2 min at 94 °C, followed by 35 cycles of 94 °C for 1 min, 60 °C for 1 min, and 72 °C for 2 min, ending with a final elongation at 72 °C for 7 min. The amplified cDNAs were cloned into the pGEM-T Easy vector (Promega, Madison, WI, USA) and sequenced by BigDye Terminator Version 3.1 cycle sequencing on ABI PRISM 377 DNA Analyzer (Applied Biosystems, Foster City, CA, USA).

The full-length cDNAs encoding the gilthead seabream vasotocin and isotocin receptors (V1a2R, V2R and ITR) were isolated from renal total RNA by RT-PCR as above using specific primers designed based on deposited sequences (GenBank accession nos. KC195974, KC960488 and KC195973, respectively). For the V1a2R forward and reverse primers were 5′-ACTATGCGCTTGTCCTGAGC-3′ and 5′-TGTGTACCTTGATGCCCAGACA-3′, respectively; for the V2R forward and reverse primers were 5′-AAAGGACACGCGTGAGAAAG-3′ and 5′-CCGTGCATGTCTTTTCAAAC-3′, respectively; and for the ITR forward and reverse primers were 5′-GACCCGGACTCTTGTGTTGT-3′ and 5′-GGGATTGCCAGGTTACTCAA-3′, respectively.

**Phylogenetic, syntenic and sequence analyses**. Using the full-length deduced proteins of the clones or exon-deduced peptides from our initial analysis[16] as tblastn queries, >1000 piscine Aqp14 peptide fragments were identified from open-source whole-genome shotgun (WGS), transcriptome shotgun (TSA) and nucleotide databases (NCBI [blast.ncbi.nlm.nih.gov], GenomeArk [vertebrategenomesproject.org] and Ensembl [ensembl.org]) and assembled contiguously. Corresponding nucleotide sequences were then retrieved from the respective DNA contigs and trimmed to match each peptide fragment, and subsequently concatenated to construct a putative cDNA for each gene. Datasets of the deduced amino acids were aligned using the L-INS-I algorithm of MAFFT v7.407[56], and subsequently converted to codon alignments using Pal2Nal[57] prior to Bayesian (Mr Bayes v3.2.2[58]) and maximum likelihood (PAUP v4b10-x86-macosx) methods as described previously[16,40]. To detect errors generated by the automated algorithms, alignments were lineage sorted according to the resulting trees and inconsistencies corrected manually using MacVector (MacVector Inc, Cambridge, UK). Bayesian phylogenetic analyses were performed on the full-length alignments (Supplementary Fig. 12) or following removal of the N-terminal exon and gapped regions containing less than three sequences (Supplementary Fig. 13). Bayesian model parameters were nucmodel = 4by4, nst = 2, rates = gamma for codon

alignments. 25 million Markov chain Monte Carlo (MCMC) generations were run with three heated and one cold chain with resulting posterior distributions examined for convergence and an effective sample size >2000 using Tracer version 1.7[59], and majority rule consensus trees summarized with a burnin of 25%. Maximum likelihood parameters used were: heuristic search optimised for parsimony; likelihood GTR model: NST = 6, with estimated parameters and siterates partitioned by codon. All trees generated were processed with Archaeopteryx[60] and rendered with Geneious (Biomatters Ltd, New Zealand). Synteny analyses were initially conducted using the Genomicus v96.01 interface[61] and subsequently via tblastn searches of WGS databases using the syntenic zebrafish genes (*ptges3b* ENSDARG00000089626, *baz2a* ENSDARG00000102974, *aqp0b* ENSDARG00000013963 and *rbm38* ENSDARG00000058818). The three-dimensional structures of the channels were built using the model leverage option in the Modeller server (modbase.compbio.ucsf.edu), based upon the human AQP4 (3GD8) template. The best scoring models were selected using the slow (Seq-Prf, PSI-BLAST) assignment method and rendered with MacPymol (pymol.org).

**Functional characterization of teleost Aqp14**. Constructs for heterologous expression in *X. laevis* oocytes were generated by subcloning the full-length gilthead seabream, Atlantic salmon and zebrafish *aqp14* cDNAs into the pT7Ts expression vector. Point mutations in the wild-type sequences were introduced using the QuikChange Lightning Site-Directed Mutagenesis Kit (Agilent Technologies). All constructs in pT7Ts were resequenced as above to assure that the correct mutations were present. The cRNA synthesis and isolation of stage V-VI oocytes were carried out as previously described[52]. Oocytes were transferred to modified Barth's solution [MBS: 88 mM NaCl, 1 mM KCl, 2.4 mM NaHCO₃, 0.82 mM MgSO₄, 0.33 mM Ca(NO₃)₂, 0.41 mM CaCl₂, 10 mM HEPES and 25 μg/ml gentamycin, pH 7.5] and injected with 50 nl of distilled water (control) or 50 nl of water solution containing 15 ng of Aqp14 cRNA. In some experiments, oocytes were previously injected with 15 ng of cRNAs encoding the V1a2R, V2R or ITR.

The $P_f$ and solute uptake of water-injected and Aqp14 expressing oocytes were determined at pH 7.5 as previously described[33,52]. The swelling assays were carried out in 10-fold diluted MBS in the presence or absence of 100 μM HgCl₂ and 5 mM β-mercaptoethanol for 20 s. The effect of PKC or PKA on oocyte $P_f$ was tested by preincubating the oocytes with 100 nM PMA for 30 min, or with 100 μM IBMX for 1 h and subsequently with 100 μM FSK for 30 min, before determination of oocyte swelling. The $P_f$ of oocytes coexpressing Aqp14 and V1a2R, V2R or ITR was measured after 3 h of exposure to 10 μM AVT (CYIQNCPRG-NH₂) or IT (CYISNCPIG-NH₂) (Peptide Protein Research Ltd., Fareham, United Kingdom), in the presence or absence of PKC and PKA inhibitors, Bim-II and H89, respectively.

Glycerol, urea and methylamine uptake were determined under isotonic conditions in the presence of 20 μCi of [1,2,3-³H]glycerol (50 mCi/mmol), [¹⁴C] urea (58 mCi/mmol), or methylamine [¹⁴C] hydrochloride (55 mCi/mmol) (American Radiolabelled Chemicals Inc., St Louis, MO), and cold glycerol, urea or methylamine at 1 mM final concentration. For H₂O₂ uptake assays, oocytes were incubated with isotonic MBS plus 0.5% DMSO and 200 μM of CM-H₂DCFDA (C6827; Life Technologies Corp.) for 1 h, and fluorescence was measured at excitation and emission wavelengths of 495 nm and 525 nm, respectively, using a multiwell plate reader (InfiniteM200, Tecan)[25].

**Statistics and reproducibility**. Data from experiments with oocytes were statistically analyzed by an unpaired Student's *t*-test using the STATGRAPHICS PLUS 4.1 software (Statistical Graphics, USA). A *P*-value < 0.05 was considered statistically significant.

**Gene expression analysis**. Extraction of total RNA from different tissues from gilthead seabream, zebrafish and Atlantic salmon (*N* = 3 for each species) and cDNA synthesis was carried out as described above. RT-PCR was performed using 1 μl of cDNA, 1 IU of Taq polymerase enzyme, and 0.5 μM of forward and reverse primers specific for each *aqp14* ortholog. For seabream, forward and reverse primers were 5′-ATGCACAGAACCAGGAAACC-3′ and 5′-CAGTCAAGCCTGATCTACACGA-3′, respectively; for zebrafish forward and reverse primers were 5′-ATGAACGCAGGTCAAGCTCT-3′ and 5′-GTCTCATGGCGCTCTGTGTA-3′, respectively; and for Atlantic salmon forward and reverse primers were 5′- GCCGGGGCTCTCTACTTC-3′ and 5′-ACTGGGGGCAAAGAAGAACT-3′, respectively. The amplification protocol was composed of an initial denaturing step for 2 min at 94 °C, followed by 35 cycles of 94 °C for 1 min, 60 °C for 1 min, and 72 °C for 2 min, ending with a final elongation at 72 °C for 7 min. PCR products were run on 1% agarose gels. For seabream and zebrafish, the reference gene was 18 s ribosomal protein (*rps18*; forward and reverse primers were 5′-ACTAAGAACGGCCATGCACCACCAC-3′ and 5′-GAATTGACGGAAGGGCACCACC-3′, respectively), whereas β-actin (*bactin*; forward and reverse primers were 5′-CCAAAGCCAACAGGGAGAAG-3′ and 5′-AGATGGGTACTGTGTGGGGTCA-3′, respectively) was the reference gene for Atlantic salmon.

**Antibody production**. An Aqp14 antiserum was raised in rabbits against a synthetic peptide corresponding to a region of the seabream Aqp14 C-terminus (amino acid residues 261–276; (TQNAMRAKQANKQENN; Agrisera AB, Vännäs,

Sweden), which is 63% and 81% identical to that of zebrafish and Atlantic salmon, respectively. The antiserum was affinity purified against the synthetic peptide.

**Immunoblotting**. Total and plasma membrane fractions of *X. laevis* oocytes were isolated as described previously[62]. Fish tissues were dissociated with a glass dounce homogenizer in ice-cold RIPA buffer containing 150 mM NaCl, 50 mM Tris-HCl, pH 8, 1% Triton X-100, 0.5% sodium deoxycholate, 1 mM EDTA, 1 mM EGTA, EDTA-free protease inhibitors (Roche Applied Science, Mannheim, Germany), 1 mM $Na_3VO4$ and 1 mM NaF, and centrifuged at 14,000 $g$ for 10 min at 4 °C. The supernatant was mixed with 2 × Laemmli sample buffer containing 200 μM di-thiothreitol, heated at 95 °C for 10 min, deep frozen in liquid nitrogen, and stored at −80 °C.

For immunoblotting, total protein extracts were denatured at 95 °C for 10 min, electrophoresed in 12% sodium dodecyl sulfate polyacrylamide gel electrophoresis (SDS-PAGE), and blotted onto Immun-Blot nitrocellulose 0.2 μm Membrane (Bio-Rad Laboratories, Hercules, CA, USA), as previously described[52]. The membranes were blocked with 5% non-fat dry milk diluted in TBST (20 mM Tris, 140 mM NaCl, 0.1% Tween; pH 8) for 1 h at room temperature, and subsequently incubated overnight at 4 °C with the seabream Aqp14 antibody (1:1000) diluted in TBST with 5% milk. Horseradish peroxidase (HRP)-coupled goat anti-rabbit IgG secondary antibodies (sc-2004; Santa Cruz Biotechnology, Dallas, TX, USA) diluted in TBST + 5% milk (1:5000) were added for 1 h at room temperature. Immunoreactive bands were revealed by the Immobilon™ Western chemiluminescent HRP substrate (MerckMillipore, Burlington, MA, USA). The specificity of the reactions were determined by incubation of duplicated membranes with the antiserum preabsorbed with the antigenic peptide.

**Immunofluorescence microscopy**. Tissue biopsies were fixed in 4% paraf-ormaldehyde in PBS (137 mM NaCl, 2.7 mM KCl, 100 mM $Na_2HPO_4$, 2 mM $KH_2PO_4$, pH 7.4) for 6 h and then washed, dehydrated and embedded in paraffin as described[52]. Sections (7 μm) were attached to UltraStick/UltraFrost Adhesion slides (Electron Microscopy Sciences, USA) and were rehydrated and then permeabilized with 0.2% Triton X 100 in PBS during 10 min before blocking with 5% normal goat serum and 0.1% bovine serum albumin for 1 h. Immunofluorescence was carried out using the anti-Aqp14 antiserum diluted in PBS (1:500, overnight, 4 °C). Antigen-antibody reaction was revealed by a sheep Cy3-coupled anti-rabbit IgG antibody (1:1000) (Sigma–Aldrich C2306) for 1 h at room temperature. The membranes and extracellular matrix were counterstained with Wheat Germ Agglutinin, Alexa Fluor® 647 Conjugate (WGA, 1:3000, Life Technologies Corp., W32466) diluted in PBS for 10 min, whereas the nuclei were counterstained with 4′,6-diamidino-2-phenylindole (DAPI, Sigma–Aldrich D9564) at 1:3000 in PBS for 3 min. The sections were washed and mounted with fluoromount aqueous anti-fading medium, and examined and photographed with a Zeiss Axio Imager Z1/ApoTome fluorescence microscope (Carl Zeiss Corp., Belcodène, France). The Aqp14-Cy3 signal was acquired in red, the DAPI in blue, and the WGA in green. Images from negative control sections were taken with the same fluorescence intensity and exposure times than those used for the positives.

**Reporting summary**. Further information on research design is available in the Nature Research Reporting Summary linked to this article.

## Data availability

The nucleotide sequences corresponding to the *aqp14* cDNAs from gilthead seabream, zebrafish and Atlantic salmon were deposited in GenBank with accession numbers MK883753, MK883754 and MK883755, respectively. All data generated or analysed during this study are included in this published article (and its supplementary information files).

## Code availability

Mr Bayes software used for phylogenetic inference is available at https://github.com/NBISweden/MrBayes/releases. PAUP software used for phylogenetic inference is available at https://paup.phylosolutions.com. MAFFT used for multiple sequence alignments is available at https://mafft.cbrc.jp/alignment/software.

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

## Acknowledgements

We thank Prof. Luca Bargelloni for early access to the gilthead seabream complete genome. This work was supported by grants from the Norwegian Research Council (Grants no. 254872/E40 and 294768/E40 to R.N.F.) and the Spanish Ministry of Economy, Industry and Competitiveness (MINECO) (Grant no. AGL2016-76802-R to J.C). Participation of F.C. and A.F. was funded, respectively, by a "Ramon y Cajal" contract (RYC-2015-17103) and a predoctoral grant (BES-2014-068745) from Spanish MINECO.

## Author contributions

R.N.F. and J.C. designed the research; F.C., O.Y., A.F. and R.N.F. performed the research; P.G.F. contributed with animal samples; F.C., O.Y., R.N.F. and J.C. analyzed the data; and F.C., R.N.F. and J.C. wrote the manuscript.

## Competing interests

The authors declare no competing interests.
