## [Peer Review File · Communications Biology]

Reviewers' comments:

Reviewer #1 (Remarks to the Author):

This paper systematically analyzes the aqp14 family, and the expression and function of aqp14 in several fishes, and its regulatory mechanism. The results make sense.

In BACKGROUND, it is better to focus on analyzing the evolution and function of the aqp family and suggest the limitations of the current study. The problem of the genome annotation should be put into the DISCUSSION section.

In results, the author should describe the results briefly instead of discussing and comparing (line 136, 334, 147 et al.)

Many aqps are widely expressed in many organs of fish, how to guaranteed the specificity of aqp antibodies?

The DISCUSSION part is weak. Could you review the current data of functions of the aqu family in the form of a table, which is easily understood.

Reviewer #2 (Remarks to the Author):

In this manuscript, Chauvigné et al. perform a functional characterization of the recently discovered aquaporin Aqp14 by studying its phylogenetic distribution, structure, expression patterns, regulation, and physiological roles. To do this, the authors employ a wide range of methodologies including sequence analysis, phylogeny inference, cloning, physiological assays, and immunofluorescence microscopy among others. The strength of this contribution is really this integrative approach to understand the function and evolution of a newly identified gene. The methods are mostly adequate as far as I can tell and the results are robust (but see below my concerns on phylogenetic inference and expression patterns). This is a very relevant contribution to the field of aquaporin research and therefore I would advise for its publication, provided that the authors adequately respond the following concerns.

Major points

Phylogeny in Fig. 1A. This phylogeny frames the position of aqp14 in relation with other members of the family. It seems odd that this has been done only including platypus and zebrafish. Why only two species? And why these and not others? I do not see why this is done "to illustrate the phylogenetic interrelationships between mammalian and piscine aquaporins". Why is a mammal-fish comparison relevant? In my opinion, a phylogeny with broader taxon sampling is the only way to clarify the phylogenetic affinities of aqp14. Fig. 1A also lacks any support values, which are essential to understand the position of aqp14 in relationship with other aquaporins. These questions seem even more relevant given that the position of aqp14 seems to differ from that reported in a previous publication (Finn et al. 2014 PLoS ONE).

Regarding the position of aqp14 within the aquaporin family, I missed information on the most likely origin of aqp14 by duplication. In this case, not only the phylogeny but synteny analysis would be relevant. The authors found aqp14 to occur in the proximity of aqp0 in most species but they did not explain how this synteny is related to the duplication history of aqp14 (and possibly aqp0)?

A second major concern is the sample size used for expression analyses. As far as I understand, the

authors used three individual: one for each of the three studied species. This made me wonder if the reported expression patterns can be really considered representative of the species, particularly for those with different expression patterns among species. I think more individuals per species would be necessary to make bold statements about the expression patterns.

Some aspects in the Methods are poorly described. First, the methods used for the generation of Fig. 1A are not provided. Other phylogenetic analyses lack the detail that is required for repeatability. Please, provide the most basic parameters for ML and Bayesian inference (model and its selection strategy, support values and number of replicates, heuristic algorithm). Also, in L507-ff the authors say that they manually corrected the alignments "to detect errors generated by automatic algorithms". Which algorithms should this be? TBLASTN? Phylogenetic inference? What kind of "inconsistencies" were observed? In L459: "aqp14 genes were identified by screening the corresponding genomes", screened how? Lastly, to ensure repeatability, the final alignments should be provided as supplementary material or even better deposited into a dedicated public repository such as Dryad or FigShare.

In the introduction, I really liked the historical perspective of the aquaporin discovery (L73-ff). However, I missed mentions to previous efforts that studied the phylogeny of aquaporins and its distribution over a broad range of species and not focused on specific species or groups like plants or vertebrates (e.g. Zardoya 2005 Biol. Cell., Abascal et al. 2014 Biochim et Biophys Acta, Finn & Cerdà 2015 Biol Bull).

Several terms and expressions used in the manuscript are evolutionarily incorrect. Expressions such as "oldest prothoterian order" (L89), "traceable to basal metazoan or parazoan lineages", and similar (L93, L184) are unnecessary and can give a wrong picture of evolution as a "ladder of progress". The expression "more basal" should not be used to refer to extant species (L159). In L169, please, modify "aqp14 likely arose with the dawn of vertebrate in agnathan fishes" to "aqp14 likely arose in the ancestor of vertebrates". This sentence makes it sound as if all other vertebrates arose from extant agnathan fish. Even if this ancestor was likely jawless, Agnatha is normally used to refer to extant taxa (lampreys and myxines). Referring to a existing molecule as "amongst the most primitive" is also incorrect (L338).

In a related note, the authors consider aqp14 as "basal members of the classical aquaporin grade" (abstract, L370)? The term basal refers to the branching time of a lineage with respect to its sister taxa (by definition, monophyletic) and a polyphyletic "grade" does not include all the descendants. Therefore, it sounds odd to use "basal" together with "grade". In addition, it would be great if the authors are more explicit about why aqp14 is considered a member of the "classical aquaporins" (see also L399). Its position as sister to all other classical aquaporins does not imply so, and thus it must be for the function. But this is never explicit.

L371: "The current evidence that hagfishes lack the aqp14 channel is consistent with the absence of aqp0 paralog in these jawless craniates" Why should the absence of aqp14 and aqp0 be consistent? Despite the conserved synteny, their phylogenetic positions do not suggest a relationship through a recent duplication event between these two paralogs. This links with the lack of information on the history of aqp14 by duplication mentioned above.

L414-ff: This is very interesting. Could the authors speculate how aqp14 might have a role in metabolite and nitrogen metabolism?

Minor points

Please, be consistent in the formatting of gene names: aqp14, Aqp14, AQP14, etc.

Reverse-transcriptase PCR is used both to amplify aqp14 sequences from mRNA as well as to identify expression patterns in different tissues. It would be good to specify the full name and not only RT-PCR, since this can be confused with real-time PCR used for quantitative expression assessments (often abbreviated RT-PCR too).

L144: Can PKA and PKC be considered widely-known abbreviations? I suggest something like "to activate kinase transduction pathways (PKA and PKC)".

L137: I think this sentence deserves clarification "we assembled each transcript de novo from the genome based upon the manual identification of 67 exons".

L124 "to assemble >1000 exons into full-length aqp14 transcripts" How many resulting aqp14 transcripts and of those, how many were full-length?

L146: cartilaginous rather than shark-related

L153: 2/3 better as %?; Also, 2/3 of the known fish orders, not of all fishes.

L163-166: "Maximum likelihood analyses confirmed this anomaly (data not shown)...": This is hardly surprising: the authors use ML and not a distance method and therefore higher sequence similarity does not necessarily correspond to phylogenetic relationships. I would suggest to remove this sentence.

L170: amino acid differences, not substitutions

L175: caecilians

L178: What do you mean with "are fractioned into pseudogenes in eutherian mammals"?

L179: fish-specific whole genome duplication

L192: I think it should be "tetraploid teleost four paralogs". Also, numbers 1-9 should probably be written (see also 5th, L409 etc.).

L216: I think these should be defined in the results: DrAqp14, SaAqp14, SsAqp14

L224: "These data suggest..." should probably be moved to the discussion or removed.

L232: "In each species..." should this be in all species?

L243: "with less intensity to that of SaAqp14" Is this correctly phrased? DrAqp14 seems to be less intense to me in Fig 3D.

L290: prompted

L321: Does the data "suggest" or "show"? Are authors not sure of their results or what is the source of uncertainty? Suggest is also used in a few other places and this should be checked.

L376: aqp0 instead of aqp01?

L384: "unique to tetrapods, having only evolved in sarcopterygian animals" This is contradictory; is it unique to tetrapods or to sarcopterygians?

L420: do the author refer to aqp0 or aqp14 pseudogenes, or both? Please, clarify.

L440-ff: Please, provide the scientific name of the used fish species too.

L448-449: unclear sentence, please rephrase.

L464: space between sites and were

L501: multiple

L646: Competitiveness (MINECO)

References: check italics (e.g. ref 14) and journal abbreviation and formatting (refs 16, 42, 47)

Fig. 2C. What is hAQP1 and hAQP3?

Reviewer #3 (Remarks to the Author):

COMMSBIO-19-1001

Overview

I enjoyed reading this manuscript and was impressed by its comprehensive and integrative approach. The study nicely combines phylogenetics and comparative genomics with cross-species molecular biology and physiology in order to advance our understanding of the evolution and function of Aqp14 in vertebrates, particularly fishes. In this respect, the reported findings are novel, and go beyond what is normal for this type of paper, and for that I applaud the authors. I feel the results will be of broad interest to those working on fish physiology (especially aquaporin biology) and evolution. In addition, the paper is well-written and convincing in most sections.

I must point out that I am not an expert in the physiological assays performed, but the methods are well-written and come from a group with a strong track record in this area of work, so I feel confident in the quality of the data.

Overall, I am supportive of publication after a minor revision to improve the presentation and clarity, along with correcting several typo and grammar issues.

Specific Comments

1. The upfront premise is that genome biology is underpinned by reliable annotation of gene functions, but most databases fail to achieve that goal via automated annotation. I could not agree more with this notion. Nonetheless, I find it an unusual premise to build the entire paper upon (i.e. first paragraph of abstract and intro), given that only a single aquaporin gene family member has been characterized, and the fact that the improved knowledge gained in the study is unlikely to feedback into the same databases in the short or medium term. One possible solution is to remove such a

strong emphasis on this point upfront in the paper, and incorporate it into a discussion point.

2. Page 9, Line 262. While it was good that the authors tested for the specificity to Aqp14 in the antiserum using the immunizing peptide, I wondered why they did not attempt to sequence the protein band/s using mass spectrometry to be certain of the identity of the proteins detected.

3. Page 10 – line 306-325. I really like the approach taken to mutate the conserved putative phosphorylation residues (Thr 241, Ser 257 and Thr 260). It clearly generated non-trivial insights. However, I wondered whether these residues act synergistically, and whether the authors considered performing mutations of distinct pairs (or all three) of each residue? Perhaps this masks the lack of effect on permeability observed for Thr 260? I consider this a non-essential suggestion.

4. phylogenetic analysis – though clearly performed appropriately, the authors must provide the sequence alignment used, along with the database versions and or accessions for each sequence used where possible. They should also provide the nucleotide substitution model used, and perhaps state the length of the codon alignments used in the analysis in the text/Figure 1 legend.

Minor points and corrections

Page 3 – line 61: “assigned to linkage groups (scaffolds/ ...). LGs and scaffolds are not interchangeable as suggested; scaffolds would normally be anchored and orientated into LGs.

Page 3 – line 76-80: can this long sentence be made shorter or broken up? It’s a lot to digest for the non-specialist.

Page 4 – line 91-99 – same point: can this long sentence be made shorter or broken up?

Page 4 – line 112: remove “which”

Page 4 – line 118 – “amongst true bony fishes (“Teleostei”); this is not right for me. True bony fishes = Osteichthyes. I suspect what is meant is “amongst true ray-finned fishes (“Actinopterygii”) as the point being made is presumably true for holosteans and non-teleosts? If teleosts is meant, the wording should be something like “Within the true bony fishes (Osteichthyes), the teleost group has evolved ... etc.”

Page 5 – line 123-125: “to assembly >1000 exons into full-length ...” it would be useful here to end this by saying “... representing X species”

Page 5 - Results: Phylogeny section. The rationale for selection of platypus is not the clearest and should be strengthened or at least elaborated

Page 5 – line 140: “platypus encodes” should be “platypus genome encodes”

Page 6 – line 171: “>500 millions of years” is more ancient for the given divergence time according to most studies I am aware of. Please confirm this is what was meant.

Page 6 – line 182: Salmonids are not tetraploid, they are paleotetraploid and most of the genome is inherited in a fully diploid mode. If they were truly tetraploid, there would be no duplicated genes to detect (this is due to the WGD being an autotetraploidization). I suggest changing to something more neutral e.g. “... were detected in the genomes of species having known WGD events, including cyprinids and salmonids” or words to that effect. Similar point for page 7 lines 192 and 194. Citations

might be given for the different WGD events also.

Page 9 – line 290: typo -> “prompted” should be “prompted”

Page 10 – line 219: “The same results are obtained when oocytes express”; should be past tense: “The same results were obtained when oocytes expressed”

Page 11 – line 336 “by the environmental” change to “by environmental”

Page 13 – line 414 – typo: “channels. are” ?

Page 14 – line 445: “for tissues collection” should be “for tissue collection”; line 446 “weighted” should be “weighed”

Page 14 – “Fish and chemical” – the size of fish for different studies should be given as ontogeny may matter to the signals demonstrated

Page 15 - line 471: “seabrean and” typo; change to “seabream and”

Page 16 – line 514: “probability distributions” should be “posterior distributions”; confirmation that the Effective Sample Size was sufficient should be given, at least for the key parameters; you might also cite: <https://academic.oup.com/sysbio/article/67/5/901/4989127>

Page 16 – line 528-529; the species used in these analyses should be explicitly listed

Page 17 – line 542: “during 20s” should be “for 20s” ?

Page 18 – 565: should “1 IU” be “1U” ?

Page 26 – Fig. 1 legend should provide substitution model and perhaps refer to where the reader can obtain the sequence alignment

Response to Reviewer 1

Key details:

1. “In BACKGROUND, it is better to focus on analyzing the evolution and function of the aqp family and suggest the limitations of the current study. The problem of the genome annotation should be put into the DISCUSSION section.
 - We have removed the problem of genome annotation from the introduction.
2. In results, the author should describe the results briefly instead of discussing and comparing (line 136, 334, 147 et al.)
 - In the referenced lines we place the results in context and consider this beneficial to the reader.
3. Many aqps are widely expressed in many organs of fish, how to guaranteed the specificity of aqp antibodies?
 - This is now specifically addressed in new experiments to demonstrate the expression of the aquaporin channels in increased numbers of individuals for each of the three species investigated. The new data are shown in Additional file 1: Fig. S3
4. The DISCUSSION part is weak. Could you review the current data of functions of the aqu family in the form of a table, which is easily understood.”
 - We have recently published two reviews (Finn and Cerdà, 2015; Finn and Cerdà, 2018) that include current data for functions of the aquaporin family of the form of tables. These data include organisms from all domains of life and are cited in the present manuscript. We therefore consider it superfluous to repeat the tables here.

Response to Reviewer 2

1. “Phylogeny in Fig. 1A. This phylogeny frames the position of aqp14 in relation with other members of the family. It seems odd that this has been done only including platypus and zebrafish. Why only two species? And why these and not others? I do not see why this is done “to illustrate the phylogenetic interrelationships between mammalian and piscine aquaporins”. Why is a mammal-fish comparison relevant? In my opinion, a phylogeny with broader taxon sampling is the only way to clarify the phylogenetic affinities of aqp14. Fig. 1A also lacks any support values, which are

essential to understand the position of aqp14 in relationship with other aquaporins. These questions seem even more relevant given that the position of aqp14 seems to differ from that reported in a previous publication (Finn et al. 2014 PLoS ONE).”

- We have redrafted this text (see lines 111-121 in the revised ms) to explain the choice as follows: “To illustrate the phylogenetic position of AQP14 in relation to the other major aquaporin subfamilies in vertebrates, we initially assembled the complete set of full-length transcripts from the prototherian platypus (*Ornithorhynchus anatinus*) and analysed them together with the established genomic repertoire of the zebrafish (*Danio rerio*) (Tingaud-Sequeira et al., 2010). We selected the platypus, since we have previously shown that it’s genome also retains the *AQP13* ortholog found in Amphibia, and each of the other subfamilies reported for tetrapods, except AQP10. Thus, with the exception of *AQP16*, which is an AQP8-type channel identified in amphibians, turtles and crocodylians, the platypus and zebrafish display each of the major divisions of aquaporin subfamilies (AQP0-15) that are representative for the two major lineages of vertebrate (Sarcopterygii and Actinopterygii) with the highest aquaporin gene copy numbers”
- We have already conducted molecular phylogenies with broader taxon sampling in the paper highlighted by the reviewer (Finn et al., 2014, PMID: 25426855), and a separate paper (Finn et al., 2015, PMID: 26183829), which demonstrate the interrelationships between the vertebrate aquaporin subfamilies, and further shows the multiplicity of the classical grade of water channels in cnidarian, protostome and deuterostome organisms.
- We agree that the annotation of the tree shown in Fig.1A was not complete. This was an oversight. The tree is mid-point rooted, and the position of Aqp14 is consistent with our previous analysis (Finn et al., 2014, PMID: 25426855). We have updated the figure to include Bayesian posterior probabilities at each node, and corrected the figure legend to include the details of the phylogenetic inference used.

2. “Regarding the position of *aqp14* within the aquaporin family, I missed information on the most likely origin of *aqp14* by duplication. In this case, not only the phylogeny but synteny analysis would be relevant. The authors found *aqp14* to occur in the proximity of *aqp0* in most species but they did not explain how this synteny is related to the duplication history of *aqp14* (and possibly *aqp0*)?”
 - As in the previous comment, this aspect is fully addressed in our previous analysis (Finn et al., 2014, PMID: 25426855). In the present manuscript we also show that the synteny of the *aqp14* genes upstream of *aqp0* is fully consistent with our previous study.
3. “A second major concern is the sample size used for expression analyses. As far as I understand, the authors used three individual: one for each of the three studied species. This made me wonder if the reported expression patterns can be really considered representative of the species, particularly for those with different expression patterns among species. I think more individuals per species would be necessary to make bold statements about the expression patterns.”
 - We have addressed this point with new experiments as explained above to the editor. The new data are shown in Additional file 1: Fig. S2.
4. “Some aspects in the Methods are poorly described. First, the methods used for the generation of Fig. 1A are not provided. Other phylogenetic analyses lack the detail that is required for repeatability. Please, provide the most basic parameters for ML and Bayesian inference (model and its selection strategy, support values and number of replicates, heuristic algorithm). Also, in L507-ff the authors say that they manually corrected the alignments “to detect errors generated by automatic algorithms”. Which algorithms should this be? TBLASTN? Phylogenetic inference? What kind of “inconsistencies” were observed? In L459: “*aqp14* genes were identified by screening the corresponding genomes”, screened how? Lastly, to ensure repeatability, the final alignments should be provided as supplementary material or even better deposited into a dedicated public repository such as Dryad or FigShare.”
 - We agree that the methods used for the generation of Fig. 1A were omitted. This was an oversight and is corrected in the revised version (see legend to Figure 1). We further provide more details of the phylogenetic inference used (see materials and methods, lines 514-520).
 - We manually corrected errors generated by the mafft alignment software that erroneously misaligns some amino acids.

- In reference to line459, *aqp14* genes were identified as explained in the materials and methods section using tblastn searches of transcriptomes and genomes.
 - A full and updated list of accession numbers included in Supplementary data (see Additional file 1: Fig. S1 and legend to Fig. S8). The codon alignments are also included in Additional file 1 (see Additional file 1: Figs S9 and S10)
- a5. “In the introduction, I really liked the historical perspective of the aquaporin discovery (L73-ff). However, I missed mentions to previous efforts that studied the phylogeny of aquaporins and its distribution over a broad range of species and not focused on specific species or groups like plants or vertebrates (e.g. Zardoya 2005 Biol. Cell., Abascal et al. 2014 Biochim et Biophys Acta, Finn & Cerdà 2015 Biol Bull).”
- This is now addressed in the revised version, with the suggested citations and others now included (see lines 61-62).
6. “Several terms and expressions used in the manuscript are evolutionarily incorrect. Expressions such as “oldest prothoterian order” (L89), “traceable to basal metazoan or parazoan lineages”, and similar (L93, L184) are unnecessary and can give a wrong picture of evolution as a “ladder of progress”. The expression “more basal” should not be used to refer to extant species (L159). In L169, please, modify “*aqp14* likely arose with the dawn of vertebrate in agnathan fishes” to “*aqp14* likely arose in the ancestor of vertebrates”. This sentence makes it sound as if all other vertebrates arose from extant agnathan fish. Even if this ancestor was likely jawless, Agnatha is normally used to refer to extant taxa (lampreys and myxines). Referring to a existing molecule as “amongst the most primitive” is also incorrect (L338).”
- We accept that some of the terms used may be semantically misinterpreted and have therefore corrected them.
7. “In a related note, the authors consider *aqp14* as “basal members of the classical aquaporin grade” (abstract, L370)? The term basal refers to the branching time of a lineage with respect to its sister taxa (by definition, monophyletic) and a polyphyletic “grade” does not include all the descendants. Therefore, it sounds odd to use “basal” together with “grade”. In addition, it would be great if the authors are more explicit about why *aqp14* is considered a member of the “classical aquaporins” (see also L399). Its position as sister to all other classical aquaporins does not imply so, and thus it must be for the function. But this is never explicit.”
- As noted above, we have previously demonstrated that *aqp14* phylogenetically clusters as a classical aquaporin as a sister branch to deuterostome and protostome

aqp4, which can be traced to basal metazoan lineages (Finn et al., 2014, PMID: 25426855; Finn et al., 2015, PMID: 26183829). Since the basal metazoan, deuterostome and protostome lineages retain multiple gene copies, we do not propose to have discovered all of the descendent genes, and therefore consider the classical aquaporins a polyphyletic grade. We have now clarified this in the discussion (see lines 364-369).

8. “L371: “The current evidence that hagfishes lack the aqp14 channel is consistent with the absence of aqp0 paralog in these jawless craniates” Why should the absence of aqp14 and aqp0 be consistent? Despite the conserved synteny, their phylogenetic positions do not suggest a relationship through a recent duplication event between these two paralogs. This links with the lack of information on the history of aqp14 by duplication mentioned above.”
 - It is consistent since that region of the genome is absent in hagfishes. We have now clarified this in the text (see lines 371-373)
9. “L414-ff: This is very interesting. Could the authors speculate how aqp14 might have a role in metabolite and nitrogen metabolism?”
 - We agree that it is interesting, but prefer in the future to investigate the roles experimentally, rather than speculate here.

Minor points

1. “Please, be consistent in the formatting of gene names: aqp14, Aqp14, AQP14, etc..”
 - We use the HUGO and Zebrafish nomenclature conventions of capitalised symbols for tetrapods, smallcase for non-tetrapods, and italicised for nucleotides.
2. “Reverse-transcriptase PCR is used both to amplify aqp14 sequences from mRNA as well as to identify expression patterns in different tissues. It would be good to specify the full name and not only RT-PCR, since this can be confused with real-time PCR used for quantitative expression assessments (often abbreviated RT-PCR too).”
 - Reverse-transcriptase PCR (RT-PCR) is defined in the text (see line 219).
3. “L44: Can PKA and PKC be considered widely-known abbreviations? I suggest something like “to activate kinase transduction pathways (PKA and PKC)”.”
 - Agreed and corrected.
4. “L137: I think this sentence deserves clarification “we assembled each transcript de novo from the genome based upon the manual identification of 67 exons”.”
 - We have added a sentence to refer the reader to the methods section.

5. "L124 "to assemble >1000 exons into full-length aqp14 transcripts" How many resulting aqp14 transcripts and of those, how many were full-length?"
 - This is now specified (see line 101).
6. "L146: cartilaginous rather than shark-related"
 - Agreed and corrected.
7. "L153: 2/3 better as %?; Also, 2/3 of the known fish orders, not of all fishes."
 - Agreed and corrected (see lines 139-140).
8. "L163-166: "Maximum likelihood analyses confirmed this anomaly (data not shown)...": This is hardly surprising: the authors use ML and not a distance method and therefore higher sequence similarity does not necessarily correspond to phylogenetic relationships. I would suggest to remove this sentence."
 - Sentence removed.
9. "L170: amino acid differences, not substitutions."
 - Agreed and corrected.
10. "L175: caecilians."
 - corrected.
11. "L178: What do you mean with "are fractioned into pseudogenes in eutherian mammals"?."
 - The term fractionated means split into smaller fragments, e.g. Langham et al., (2004) Genomic duplication, fractionation and the origin of regulatory novelty. Genetics 166, 935-945.
12. "L179: fish-specific whole genome duplication"
 - corrected.
13. "L192: I think it should be "tetraploid teleost four paralogs". Also, numbers 1-9 should probably be written (see also 5th, L409 etc)."
 - corrected.
14. "L216: I think these should be defined in the results: DrAqp14, SaAqp14, SsAqp14"
 - Now defined.
15. "L224: "These data suggest..." should probably be moved to the discussion or removed."
 - We do not think that this is necessary.

16. "L232: "In each species..." should this be in all species?."
 - No, since "all" could be interpreted as species that we have not tested.
17. "L243: "with less intensity to that of SaAqp14" Is this correctly phrased? DrAqp14 seems to be less intense to me in Fig 3D."
 - Exactly, yes this is correctly phrased.
18. "L290: prompted."
 - corrected.
19. "L321: Does the data "suggest" or "show"? Are authors not sure of their results or what is the source of uncertainty? Suggest is also used in a few other places and this should be checked."
 - This is a style point, and we prefer to use suggest.
20. "L376: aqp0 instead of aqp01?"
 - aqp01 is correct (see Finn et al., 2014; PMID: 25426855).
20. "L384: "unique to tetrapods, having only evolved in sarcopterygian animals" This is contradictory; is it unique to tetrapods or to sarcopterygians?"
 - corrected.
21. "L420: do the author refer to aqp0 or aqp14 pseudogenes, or both? Please, clarify."
 - AQP14, now clarified.
22. "L440-ff: Please, provide the scientific name of the used fish species too."
 - Now provided.
23. "L448-449: unclear sentence, please rephrase."
 - Sentence rephrased.
24. "L464: space between sites and were."
 - corrected.
25. "L501: multiple."
 - corrected.
26. "L646: Competitiveness (MINECO)."
 - corrected.
27. "References: check italics (e.g. ref 14) and journal abbreviation and formatting (refs 16, 42, 47)"
 - checked and formatted according to the journal requirements.

28. “Fig. 2C. What is hAQP1 and hAQP3?”

- corrected.

Response to Reviewer 3

Key details:

1. “The upfront premise is that genome biology is underpinned by reliable annotation of gene functions, but most databases fail to achieve that goal via automated annotation. I could not agree more with this notion. Nonetheless, I find it an unusual premise to build the entire paper upon (i.e. first paragraph of abstract and intro), given that only a single aquaporin gene family member has been characterized, and the fact that the improved knowledge gained in the study is unlikely to feedback into the same databases in the short or medium term. One possible solution is to remove such a strong emphasis on this point upfront in the paper, and incorporate it into a discussion point.
 - We concur and have modified the text accordingly.
2. Page 9, Line 262. While it was good that the authors tested for the specificity to Aqp14 in the antiserum using the immunizing peptide, I wondered why they did not attempt to sequence the protein band/s using mass spectrometry to be certain of the identity of the proteins detected.
 - The specificity of the antiserum has been addressed (see Additional file 1: Fig. S3). Mass spectrometry may be confusing as many different proteins are present in the gel slide that corresponds to the immunoreactive band. We believe that the additional experiments carried out confirm the specificity of the antiserum.
3. Page 10 – line 306-325. I really like the approach taken to mutate the conserved putative phosphorylation residues (Thr 241, Ser 257 and Thr 260). It clearly generated non-trivial insights. However, I wondered whether these residues act synergistically, and whether the authors considered performing mutations of distinct pairs (or all three) of each residue? Perhaps this masks the lack of effect on permeability observed for Thr 260? I consider this a non-essential suggestion.
 - A synergistic effect of two residues on Aqp14 trafficking may be possible, but as the reviewer mentioned, these experiments go beyond the scope of the present work.

4. phylogenetic analysis – though clearly performed appropriately, the authors must provide the sequence alignment used, along with the database versions and or accessions for each sequence used where possible. They should also provide the nucleotide substitution model used, and perhaps state the length of the codon alignments used in the analysis in the text/Figure 1 legend.”
 - We provide new accession numbers for the sequences used and the alignments in Additional file 1: Figs S1, S8, S9 and S10, respectively.

Minor points

1. “Page 3 – line 61: “assigned to linkage groups (scaffolds/ ...). LGs and scaffolds are not interchangeable as suggested; scaffolds would normally be anchored and orientated into LGs.”
 - Sentence removed.
2. “Page 3 – line 76-80: can this long sentence be made shorter or broken up? It’s a lot to digest for the non-specialist.”
 - Sentence rephrased.
3. “Page 4 – line 91-99 – same point: can this long sentence be made shorter or broken up?”
 - Sentence rephrased.
4. “Page 4 – line 112: remove “which””
 - “which” refers to the piscine genomes, and therefore remains.
5. “Page 4 – line 118 – “amongst true bony fishes (“Teleostei”); this is not right for me. True bony fishes = Osteichthyes. I suspect what is meant is “amongst true ray-finned fishes (“Actinopterygii”) as the point being made is presumably true for holosteans and nonteleosts? If teleosts is meant, the wording should be something like “Within the true bony fishes (Osteichthyes), the teleost group has evolved ... etc.””
 - Sentence updated to reflect this point.
6. “Page 5 – line 123-125: “to assembly >1000 exons into full-length ...” it would be useful here to end this by saying “... representing X species”
 - Sentence updated to reflect the number of full-length and partial sequences assembled (see line 101).
7. “Page 5 - Results: Phylogeny section. The rationale for selection of platypus is not the clearest and should be strengthened or at least elaborate”
 - This is now clarified as explained in point 1 to reviewer 2.

8. "Page 5 – line 140: "platypus encodes" should be "platypus genome encodes""
 - corrected.
9. "Page 6 – line 171:" >500 millions of years" is more ancient for the given divergence time according to most studies I am aware of. Please confirm this is what was meant."
 - This is correct as cited in the text.
10. "Page 6 – line 182: Salmonids are not tetraploid, they are paleotetraploid and most of the genome is inherited in a fully diploid mode. If they were truly tetraploid, there would be no duplicated genes to detect (this is due to the WGD being an autotetraploidization). I suggest changing to something more neutral e.g. "... were detected in the genomes of species having known WGD events, including cyprinids and salmonids" or words to that effect. Similar point for page 7 lines 192 and 194. Citations might be given for the different WGD events also."
 - Sentence corrected to reflect this point.
11. "Page 9 – line 290: typo -> "prompted" should be "prompted""
 - corrected.
12. "Page 10 – line 219: "The same results are obtained when oocytes express"; should be past tense: "The same results were obtained when oocytes expressed""
 - corrected.
13. "Page 11 – line 336 "by the environmental" change to "by environmental""
 - corrected.
14. "Page 13 – line 414 – typo: "channels. are" ?"
 - corrected.
15. "Page 14 – line 445: "for tissues collection" should be "for tissue collection"; line 446 "weighted" should be "weighed""
 - corrected.
16. "Page 14 – "Fish and chemical" – the size of fish for different studies should be given as ontogeny may matter to the signals demonstrated"
 - Fish size now provided in the text.
17. "Page 15 - line 471: "seabrean and" typo; change to "seabream and""
 - corrected.

18. “Page 16 – line 514: “probability distributions” should be “posterior distributions”; confirmation that the Effective Sample Size was sufficient should be given, at least for the key parameters; you might also cite:
[https://academic.oup.com/sysbio/article/67/5/901/4989127.](https://academic.oup.com/sysbio/article/67/5/901/4989127)”
 - Text updated to state “posterior distributions” and the effective sample size, and the above paper is now cited.
19. “Page 16 – line 528-529; the species used in these analyses should be explicitly listed.”
 - Species now listed.
20. “Page 17 – line 542: “during 20s” should be “for 20s” ?”
 - corrected.
21. “Page 18 – 565: should “1 IU” be “1U” ?”
 - IU is standard for international units.
22. “Page 26 – Fig. 1 legend should provide substitution model and perhaps refer to where the reader can obtain the sequence alignment”
 - Legend updated to include phylogenetic details, and alignments provided in Additional file 1: Figs S9 and S10.

We thank the reviewers for their constructive comments and hope that the changes introduced in the present revised manuscript are satisfactory.

REVIEWERS' COMMENTS:

Reviewer #1 (Remarks to the Author):

There are more than 30,000 species of fish, and the diversity of the environment has caused fish diversity. Their genetic diversity and complexity are high, AQP is a star molecule, and relevant research is very meaningful. This paper has carried out a more detailed demonstration of AQP14 expression, cell localization and gene regulation. The work of phylogenetic analysis and cell localization is impressive. In the revised manuscript, the author made a detailed revision and reply to the reviewer's comments. The paper could provide more information to AQP researchers on the genetic data of the AQP family.

Reviewer #2 (Remarks to the Author):

The new manuscript is much improved compared to the earlier version. I enjoyed the discussion of AQP diversity and taxonomic distribution in the discussion and the transfer of the genome annotation problem to the discussion. I have mostly minor corrections, which the authors can be trusted to correct.

Although the authors corrected most inaccurate uses of evolutionary terms, there remain some to be addressed.

L70. Cnidaria and Porifera are considered "basal metazoans". This is wrong. Most other uses of basal seem fixed. The distinction between "ancient" and "modern" lineages of fishes should also be checked (L104-5), what do the authors mean?

L136-7: "AQP14 clustering next to AQP4 within the classical grade of aquaporins". This reflects a common mistake of reading trees ladder-wise. AQP14 is the sister group to all other classical aquaporins. Therefore, AQP14 is as close to AQP4 as it is to the clade containing AQP0-2,4-6,15. It is always safe to describe phylogenetic trees in terms of sister-group relationships. Please, rephrase this.

L333. What does it mean that a molecule is "primitive"? An early evolutionary origin of the family would not justify such a name. The molecules the authors refer to belong to extant species, and thus cannot be primitive.

I had previously shown concern regarding the small sample size of the expression experiments. The authors increased the sample size from 1 to 3 individuals per species (Suppl. Fig. S2), which is much better. However, the data show quite some inter-specific variability and need to be interpreted with caution. I would suggest to the authors to clearly indicate the number of individuals (biological replicates) that are used per species (e.g. in the Methods this is not explicit) and clearly note the limitations of their sample size in the Discussion.

I appreciate that the authors provide the alignments as supplementary figures. However, it would be better to provide them as text files because this format, unlike figures, makes this data usable for future research.

L98. Salt instead of salts?

L103-5. I think it should be demonstrated and confirmed. Also, what does it mean "the existence of the complete ortholog"? Does it refer to aqp14?

L115. Its

L120. Vertebrates

L212. Capitalize Ensembl

L122. This is better described as “we assembled the full CDS upon manual identification of 67 exons” or similar. Assembled the transcriptome de novo might be confused with a transcriptome approach.

L132. Why cartilaginous-related and not simply cartilaginous?

L148-52. Lower sequence identity does not directly translate into closer phylogenetic relationships when probabilistic methods are used. I have previously pointed this out despite the authors claim to have removed it, it remains in the text.

L159. I think a closing parenthesis is missing after ref. 32

L161. The authors use indistinctively Lissamphibia and Amphibia. For simplicity and accuracy, I suggest using always the former.

L179. I think the comma after salmon should be removed.

L189-ff; L409, L425, L514, L516 Numbers up to 10 are better written in text.

L321. tetrapods

L352. I think it should be “more and less, respectively”

L370. Nearly all?

L373. I am still not convinced of the authors' claim that the absence of aqp14 and aqp0 are consistent. Why should they be consistent? Cannot one gene get lost while the other one is kept in agnathans? Maybe a key point would be to understand if maybe this whole region is absent from the genomes of agnathans, or if instead the region is present (e.g. identified by synthetic genes) but both aqp0 and aqp14 are missing.

L421. Some of which retain AQP14 pseudogenes.

L503. Remove “in multiple sequence alignments”? Mafft was not applied at this point.

L811. Remove colon after zebrafish?

Fig. 1A. Posterior probabilities are in the range 0-1, not percent. All posterior probabilities below 0.9 (ideally 0.95) mean that the branches are not credible, and thus differentiating branches with posterior probabilities 0.5-0.9 is not relevant and I would suggest to remove them for simplicity. Scale bar should be expected substitutions per site.

Reviewer #3 (Remarks to the Author):

I previously reviewed this manuscript, and remain strongly supportive of its publication in a revised form; the study represents a strong contribution to the field.

Overall, I find that the authors have satisfactorily addressed the full set of comments from myself. They also seem to have addressed the comments of the other two reviewers, including by performing additional experiments and revising the written text in many places.

At this stage, I only have minor remaining comments:

- Please upload the alignments in a data format that can be used by others. I find the provision of the alignments as PDFs unsatisfactory as other could not use them in future analyses.

Typos:

- line 99: "enviroment"
- line 133: "...mammals, however the zebrafish encodes..." change to "...mammals, the zebrafish genome encodes ..."
- line 140: "Bayesian analyses" change to "Bayesian phylogenetic analyses"
- line 159: the parenthesis needs to be closed after ref 32

Finally, as feedback for future reference, the responses provided by the authors were not particularly informative, and I would have appreciated a clearer explanation of what changes were actually performed in the response letter, rather than having to dig into the manuscript to find out how the critiques were addressed.

Response to Reviewer 1

Key details:

1. “There are more than 30,000 species of fish, and the diversity of the environment has caused fish diversity. Their genetic diversity and complexity are high, AQP is a star molecule, and relevant research is very meaningful. This paper has carried out a more detailed demonstration of AQP14 expression, cell localization and gene regulation. The work of phylogenetic analysis and cell localization is impressive. In the revised manuscript, the author made a detailed revision and reply to the reviewer's comments. The paper could provide more information to AQP researchers on the genetic data of the AQP family.”
 - An overview of the vertebrate aquaporin superfamily is provided in Figure 1A. The rest of the manuscript is dedicated to the novel Aqp14 channel as outlined in the title.

Response to Reviewer 2

1. “Although the authors corrected most inaccurate uses of evolutionary terms, there remain some to be addressed.

L70. Cnidaria and Porifera are considered “basal metazoans”. This is wrong. Most other uses of basal seem fixed. The distinction between “ancient” and “modern” lineages of fishes should also be checked (L104-5), what do the authors mean?”
 - As stated in the text, we refer to the lineages as basal, not the organisms. The term basal is used to indicate lineages closer to the metazoan root, and is thus a correct usage of the adjective. The same refers to the use of “ancient” and “modern” lineages of fishes as those closer or more distant from the root, respectively. The terms are therefore maintained in the text.
2. “L136-7: “AQP14 clustering next to AQP4 within the classical grade of aquaporins”. This reflects a common mistake of reading trees ladder-wise. AQP14 is the sister group to all other classical aquaporins. Therefore, AQP14 is as close to AQP4 as it is to the clade containing AQP0-2,4-6,15. It is always safe to describe phylogenetic trees in terms of sister-group relationships. Please, rephrase this.”
 - As stated in our previous response, we have earlier shown that the Aqp14 branch is sister to the Aqp4 channels (see PMID: 25426855), which together are sister to all other vertebrate classical aquaporins. The present tree (Fig. 1A) is provided to

illustrate the phylogenetic separation of the superfamily (Aqp0 - Aqp15) into the four established grades of classical aquaporins, Aqp8-type aquaporins, aquaglyceroporins and unorthodox aquaporins. This is achieved by selecting the genomic repertoires of the platypus and zebrafish as explained in the results section. Although, the tree may superficially reflect a ladder with the two species analysed, it would not be correct to state that the Aqp14 subfamily is sister all other classical aquaporins based upon a two-species analysis. We have therefore maintained the text as “*AQP14* clustering next to *AQP4* within the classical grade of aquaporins”.

3. “L333. What does it mean that a molecule is “primitive”? An early evolutionary origin of the family would not justify such a name. The molecules the authors refer to belong to extant species, and thus cannot be primitive.”
 - The use of the adjective “primitive” in this context means amongst the first of its kind, and is therefore in accordance with the dictionary definition.
4. “I had previously shown concern regarding the small sample size of the expression experiments. The authors increased the sample size from 1 to 3 individuals per species (Suppl. Fig. S2), which is much better. However, the data show quite some inter-specific variability and need to be interpreted with caution. I would suggest to the authors to clearly indicate the number of individuals (biological replicates) that are used per species (e.g. in the Methods this is not explicit) and clearly note the limitations of their sample size in the Discussion.”
 - We agree with the reviewer that our data show that in each species, prominent mRNA *aqp14* expression is detected in the brain, lens and testis, but in other tissues it shows greater variability between the species, and in some cases between animals. This is clearly stated in the results (see lines 220-224). Our major observations are however confirmed for the translated Aqp14 proteins in Western blot and immunocytochemical experiments using a specific antibody (data are shown in Figs 3D-F, 4, 5 and Supplementary Figure S3D, E). To clarify the number of biological replicates, we have added (N=3 for each species) to the Fig. 3 legend, and in the materials and methods section (see lines 569 and 854).
5. “I appreciate that the authors provide the alignments as supplementary figures. However, it would be better to provide them as text files because this format, unlike figures, makes this data usable for future research.”
 - Text files are now provided (see Supplementary Figures 12 and 13)

6. “L98. Salt instead of salts?”
 - We use the plural “salts” here to indicate more than one type of ion.
7. “L103-5. I think it should be demonstrated and confirmed. Also, what does it mean “the existence of the complete ortholog”? Does it refer to *aqp14*?”
 - The experiments set out in the manuscript demonstrate the molecular function and neuropeptide regulation of the Aqp14 channel. Yes it refers to the complete Aqp14 ortholog, which we have now clarified in the text (see lines 103-104). We further confirm that the complete Aqp14 ortholog is present in all extant sarcopterygian lineages (see Supplementary Figure 11 for the C-terminal alignment and the legend for the accession numbers of the full-length sequences).
8. “L115. Its”
 - Corrected (see line 113)
9. “L120. Vertebrates”
 - Corrected (see line 117)
10. “L212. Capitalize Ensembl”
 - Corrected (see line 119)
11. “L122. This is better described as “we assembled the full CDS upon manual identification of 67 exons” or similar. Assembled the transcriptome de novo might be confused with a transcriptome approach.”
 - We have adjusted the text to state “full CDS”, to avoid such confusion (see line 120).
12. “L132. Why cartilaginous-related and not simply cartilaginous?”
 - Corrected (see line 129)
13. “L148-52. Lower sequence identity does not directly translate into closer phylogenetic relationships when probabilistic methods are used. I have previously pointed this out despite the authors claim to have removed it, it remains in the text.”
 - We did indeed remove the sentence “Maximum likelihood analyses confirmed this anomaly (data not shown)” in the original ms, however the sentence referring to the lower identity of the muskipper *aqp14* transcripts remains as a factual record of our observations.
14. “L159. I think a closing parenthesis is missing after ref. 32”
 - Corrected (see line 156)

15. “L161. The authors use indistinctively Lissamphibia and Amphibia. For simplicity and accuracy, I suggest using always the former.”
 - Corrected to Amphibia, which represents the class rather than the subclass (see line 158).
16. “L179. I think the comma after salmon should be removed.”
 - Comma removed (see line 176)
17. “L189-ff; L409, L425, L514, L516 Numbers up to 10 are better written in text.”
 - Corrected (see lines 186, 406, 408, 424, 512, 515)
18. “L321. tetrapods”
 - Corrected (see line 320)
19. “L352. I think it should be “more and less, respectively”
 - No the text is correct as referring to “more or less P_f ”.
20. “L370. Nearly all?”
 - Corrected (see line 369)
21. “L373. I am still not convinced of the authors’ claim that the absence of *aqp14* and *aqp0* are consistent. Why should they be consistent? Cannot one gene get lost while the other one is kept in agnathans? Maybe a key point would be to understand if maybe this whole region is absent from the genomes of agnathans, or if instead the region is present (e.g. identified by synthetic genes) but both *aqp0* and *aqp14* are missing.”
 - As stated in the text and explained previously, the region is missing in the hagfish genome. Consequently the absence of the region is consistent with the absence of both the *aqp14* and the *aqp0* orthologs.
22. “L421. Some of which retain AQP14 pseudogenes.”
 - We prefer not to adjust this text, since we refer to the lineage of eutherian mammals only.
23. “L503. Remove “in multiple sequence alignments”? Mafft was not applied at this point.”
 - Removed (see line 502)
24. “L811. Remove colon after zebrafish?”
 - Removed (see line 821)

25. "Fig. 1A. Posterior probabilities are in the range 0-1, not percent. All posterior probabilities below 0.9 (ideally 0.95) mean that the branches are not credible, and thus differentiating branches with posterior probabilities 0.5-0.9 is not relevant and I would suggest to remove them for simplicity. Scale bar should be expected substitutions per site."
- We have re-annotated the percentages as probabilities in Fig. 1. As stated in the legend, the tree is a majority rule consensus tree with all annotated nodes delineated by >0.5 posterior probability. We therefore show all of the data, which is required by the journal. The annotation for the scale bar is changed to expected substitutions per site (see lines 829-830).

Response to Reviewer 3

Key details:

"I previously reviewed this manuscript, and remain strongly supportive of its publication in a revised form; the study represents a strong contribution to the field. Overall, I find that the authors have satisfactorily addressed the full set of comments from myself. They also seem to have addressed the comments of the other two reviewers, including by performing additional experiments and revising the written text in many places.

At this stage, I only have minor remaining comments:

1. Please upload the alignments in a data format that can be used by others. I find the provision of the alignments as PDFs unsatisfactory as other could not use them in future analyses..
 - Alignments are provided as text files (Supplementary Figures 12 and 13).
2. "line 99: "enviroment""
 - Corrected (see line 93).
3. "line 133: "...mammals, however the zebrafish encodes..." change to "...mammals, the zebrafish genome encodes ..."."
 - Corrected (see line 127).
4. "line 140: "Bayesian analyses" change to "Bayesian phylogenetic analyses""
 - Corrected (see line 137).

5. “-line 159: the parenthesis needs to be closed after ref 32”
 - Corrected (see line 156).

6. “Finally, as feedback for future reference, the responses provided by the authors were not particularly informative, and I would have appreciated a clearer explanation of what changes were actually performed in the response letter, rather than having to dig into the manuscript to find out how the critiques were addressed.”
 - We have prepared a detailed response to each of the points raised by the reviewers, with line numbers reported for all of the corrections.

We thank the reviewers for their constructive comments and hope that the changes introduced in the present revised manuscript are satisfactory.